# Mapping Semantic & Syntactic Relationships with Geometric Rotation

**Michael Freenor & Lauren Alvarez**
TELUS Digital
{michael.freenor,lauren.alvarez}@telusdigital.com

## Abstract

Understanding how language and embedding models encode semantic relationships is fundamental to model interpretability. While early word embeddings exhibited intuitive vector arithmetic ("king" - "man" + "woman" = "queen"), modern high-dimensional text representations lack straightforward interpretable geometric properties. We introduce Rotor-Invariant Shift Estimation (RISE), a geometric approach that represents semantic-syntactic transformations as consistent rotational operations in embedding space, leveraging the manifold structure of modern language representations. RISE operations have the ability to operate across both languages and models without reducing performance, suggesting the existence of analogous cross-lingual geometric structure. We compare and evaluate RISE using two baseline methods, three embedding models, three datasets, and seven morphologically diverse languages in five major language groups. Our results demonstrate that RISE consistently maps discourse-level semantic-syntactic transformations with distinct grammatical features (e.g., negation and conditionality) across languages and models. This work provides the first demonstration that discourse-level semantic-syntactic transformations correspond to consistent geometric operations in multilingual embedding spaces, empirically supporting the linear representation hypothesis at the sentence level.

## 1 Introduction

Understanding how contemporary language models encode and manipulate semantic knowledge has become a central challenge in deep learning interpretability. The ability to interpret (probe) and control (steer) these internal representations is fundamental to developing trustworthy, safe AI systems. In word2vec (Mikolov et al., 2013a) and similar models, semantic relationships could be captured with simple vector arithmetic in the embedding space (i.e. the famous "king" - "man" + "woman" = "queen" analogy). This linear transparency offered both interpretability and controllability, enabling researchers to navigate semantic space through intuitive mathematical operations.

However, this clarity has largely disappeared in modern transformer-based language models. While large language models (LLMs) have achieved remarkable performance across diverse language tasks (Achiam et al., 2023; Touvron et al., 2023), their internal workings remain largely opaque (Elhage et al., 2022; Rogers et al., 2021), limiting our ability to understand, predict, and control their behavior in critical applications. Unlike the interpretable, linear directions found in static word embeddings, the geometry of modern text representations lacks the same straightforward correspondence to semantic operations. This opacity poses significant challenges for understanding how these models organize linguistic knowledge and limits our ability to interpret their behavior in principled ways.

The central challenge lies in identifying which geometric operations correspond to meaningful semantic transformations in these complex representation spaces. Current approaches often rely on task-specific *probes* (Rogers et al., 2021; Hewitt & Manning, 2019; Alain & Bengio, 2017) or *steering vectors* (Zou et al., 2023; Wang et al., 2023; Turner et al., 2023; Merullo et al., 2024; Trager et al., 2023), but lack generalizable frameworks for systematically mapping semantic relationships to geometric structure. Without such principled methods, we cannot determine whether the geometric regularities that made static word embeddings interpretable persist in modern language or embedding models, albeit in more complex forms.

We address this gap by introducing Rotor-Invariant Shift Estimation (RISE), a geometric approach that represents semantic-syntactic transformations as consistent rotational operations in embedding space, leveraging the manifold structure of modern language representations. RISE is a rotor-based alignment method that identifies cross-lingual and cross-model geometric transformations. Specifically, we demonstrate how RISE identifies three discourse-level semantic-syntactic changes (negation, conditionality, and politeness) across seven morphologically distinct languages and generalizes across three different embedding model architectures. **The goal of this study is to develop a framework for identifying discourse-level semantic-syntactic changes that correspond to consistent geometric transformations, and determine how well these transformations can be cross-lingually mapped across model architectures.** Our approach treats semantic-syntactic transformations as rotations on the unit hypersphere, where sentence embeddings reside, enabling us to align different linguistic contexts into a common geometric framework. This paper presents evidence that certain semantic-syntactic transformations exhibit generalizable geometric structure while others vary based on context-dependence, extending the linear representation hypothesis to cross-lingual discourse. We demonstrate this through empirical experiments across two baselines, three models, and seven languages – revealing that negation, conditionality, and politeness transformations can be captured as consistent rotational operations [1].

## 2 Related Work

### 2.1 Linear Representation Hypothesis

The linear representation hypothesis (LRH), or linear subspace hypothesis, has emerged as a promising theory for bridging the interpretability gap for embeddings (Mikolov et al., 2013b; Levy & Goldberg, 2014; Bolukbasi et al., 2016; Ethayarajh, 2019; Park et al., 2024; 2025). The LRH posits that semantic concepts are encoded as linear structures within embedding spaces, meaning linear algebraic operations can be used for interpretation and control (e.g., "king" - "man" + "woman" = "queen" presented by Mikolov et al. (2013b)). Park et al. (2024) formalized the LRH by unifying three distinct notions of linearity that had developed independently across the literature:

1. word2vec-like embedding differences (Arora et al., 2016; Mimno & Thompson, 2017; Ethayarajh et al., 2019; Reif et al., 2019; Li et al., 2020; Hewitt & Manning, 2019; Chen et al., 2021; Chang et al., 2022; Jiang et al., 2023; Mitchell & Lapata, 2008; Baroni & Zamparelli, 2010)

2. logistic probing (Alain & Bengio, 2017; Kim et al., 2018; Belinkov, 2022; Li et al., 2022; Geva et al., 2022; Nanda et al., 2023)

3. steering vectors (Wang et al., 2023; Turner et al., 2023; Merullo et al., 2024; Trager et al., 2023)

Park et al.'s theoretical framework addresses a critical gap by synthesizing the first formalization of what "linear representation" means (Park et al., 2024). However, while the LRH has been validated primarily within individual languages (monolingually), there remains a significant gap in understanding how semantic-syntactic transformations generalize across linguistic contexts (cross-lingually). Most existing work examines static concept encodings (Park et al., 2025; 2024) rather than dynamic semantic-syntactic transformations that reflect real-world language use. Our work is the first to extend the LRH to multilingual contexts and embedding models. Although, the linear representations we consider are geodesic arcs and not Euclidean lines.

### 2.2 Linear & Geometric Representation Techniques

The geometric foundations established by Park et al. (2024) are crucial for understanding when and why linear algebraic operations succeed in capturing semantic relationships. With traditional Euclidean geometry, it is hard to accept that arbitrary dot products or cosine similarities have semantic meaning. Moreover, Park et al. (2024) demonstrated that the choice of inner product fundamentally determines the interpretability of geometric operations, providing principled foundations for

---

[1]The link to our GitHub repository is `https://github.com/fuelix/RISE-steering`.

representation analysis. Our work builds directly on recent advances in understanding linear representations in language models (Park et al., 2024; Li et al., 2023). RISE implements a technique that respects semantic structure, similar to the geometric framework developed by Park et al. (2024). While previous work focused primarily on categorical concepts and word-level transformations, RISE extends our understanding to sentence-level, discourse-level transformations through cross-lingual and cross-model analysis using seven morphologically diverse languages.

### 2.2.1 STEERING VECTORS & EMBEDDING MODELS

The practical applications of linear representation theory have been explored through steering vector techniques. Turner et al. (2023), Liu et al. (2024), and Zou et al. (2023) demonstrated that targeted modifications to internal, latent space representations can systematically alter model behavior without parameter updates. The majority of steering vector research (Im & Li, 2025; Rimsky et al., 2023; Zou et al., 2023; Li et al., 2023) is connected to activation steering, only investigating the impact of steering vectors in the activation, hidden, and/or latent layer of an LLM. Recently, Pham & Nguyen (2024) introduced Householder Pseudo-Rotation (HPR), which addresses activation norm consistency issues in LLM behavioral modification through direction-magnitude decomposition and pseudo-rotational transformations. Building on the insight that geometric approaches outperform additive methods, our work extends geometric reasoning to semantic transformations in embedding space through Riemannian operations. To our knowledge, there is no work investigating the application of steering vectors to embedding models – only completion models. This study extends steering principles to embedding models on manifolds, not activation-level steering.

### 2.3 GENERALIZATION AND RELIABILITY CHALLENGES

Current knowledge about the generalization properties of linear representations reveals significant limitations. The taxonomy of generalization research in natural language processing (NLP) (Hupkes et al., 2023) provides a framework for evaluating robustness, but systematic applications to representation-based techniques (i.e., steering, probing, or embedding manipulation) have been limited. Recent empirical studies have revealed that steering vector effectiveness varies substantially across different inputs and contexts (Tan et al., 2024). Secondly, the relationship between local and global linearity represents a particularly critical gap in current understanding. There have been numerous demonstrations of local linear behavior within specific domains or prompt formats, but achieving global linearity (generalizable to multiple model architectures with different pre-training as required by strong versions of the LRH) remains challenging. While many studies demonstrate impressive results in controlled settings, they often fail to address the robustness needed in practical applications. This study contributes to the literature gap by presenting a robust framework for geometrically identifying discourse-level semantic-syntactic changes across typologically diverse languages and model architectures.

## 3 THEORETICAL MOTIVATION

The limitations identified in the related literature point toward a fundamental, theoretical challenge: existing approaches operate in Euclidean/linear space while modern embeddings live on curved manifolds (spherical space). This geometric mismatch may explain why steering vector research shows inconsistent cross-context performance and why linear methods struggle with robust generalization. We **hypothesize** that discourse-level semantic-syntactic transformations correspond to intrinsic geometric operations on the embedding manifold, rather than fixed directions derived from Euclidean computations. If semantic transformations can be characterized as consistent rotational operations on the unit hypersphere where embeddings reside, **this would provide theoretical support for the extension of the Linear Representation Hypothesis in curved spaces (through geodesics) and cross-lingual interpretability.** Testing this hypothesis requires robust evaluation across diverse languages and embedding architectures to determine whether geometric consistency reflects universal semantic properties or model-specific artifacts.

## 4 ROTOR-INVARIANT SHIFT ESTIMATION (RISE)

Modern sentence embeddings from multilingual encoders reside approximately on a unit hypersphere in high-dimensional space when the training objective enforces or fixes the $\ell_2$-norm constraints (Hirota et al., 2020), the embeddings are normalized to unit length (Reimers & Gurevych, 2019), or the model is designed to produce isotropic embeddings (Li et al., 2020; Ethayarajh, 2019). Local semantic transformations (e.g., negation, politeness, conditionality) can be understood as rotational displacements on this sphere. The key insight is that these displacements can be interpreted by aligning different contexts to a common geometric frame.

For any neutral sentence embedding $n \in \mathbb{S}^{d-1}$ and its semantically transformed variant $v \in \mathbb{S}^{d-1}$, we can compute an orthogonal transformation (Clifford-algebraic rotor) $R(n)$ that aligns $n$ to a canonical reference direction $e_1$. By applying this same transformation to $v$, we express the semantic change in a standardized coordinate system:

$$\xi = R(n) \, \log_n(v), \tag{1}$$

where $\log_n(v)$ denotes the Riemannian logarithm that computes the tangent vector from $n$ to $v$ on the hypersphere, and $R(n)$ aligns the tangent vector to the canonical reference direction. Normalized embeddings reside on a unit hypersphere, where geodesics define the shortest paths between points, preserving the manifold's intrinsic geometry rather than imposing Euclidean distance measures. These geodesic paths represent the natural notion of a "line" in the embedding space, as they define the shortest distance between two points on the surface. By working with geodesics, we ensure our semantic transformations are consistent with the manifold structure. To "flatten" out the curved arc to a straight vector, the Riemannian logarithmic map $\log_n(v)$ produces the vector from $n$ to $v$ on a tangent plane at $n$. By operating within the tangent space at $n$, geodesic differences can be treated as ordinary vectors.

### 4.1 THE ROTOR ALIGNMENT ALGORITHM

RISE proceeds in three steps illustrated in Figure 1:

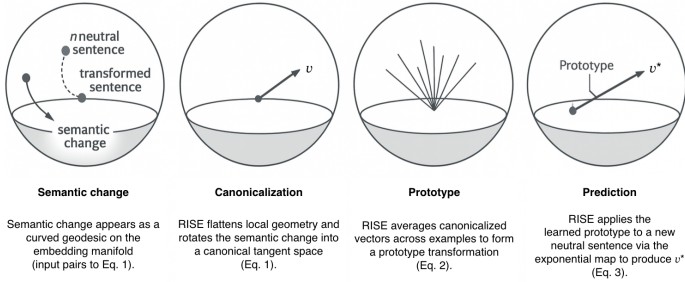

**Semantic change**

Semantic change appears as a curved geodesic on the embedding manifold (input pairs to Eq. 1).

**Canonicalization**

RISE flattens local geometry and rotates the semantic change into a canonical tangent space (Eq. 1).

**Prototype**

RISE averages canonicalized vectors across examples to form a prototype transformation (Eq. 2).

**Prediction**

RISE applies the learned prototype to a new neutral sentence via the exponential map to produce $v^*$ (Eq. 3).

Figure 1: RISE step-by-step illustration.

**Canonicalization.** For each neutral–transformed sentence pair $(n_i, v_i)$, compute a rotor $R(n_i)$ that maps $n_i$ to the reference direction $e_1$. We interpret canonicalization as controlling for the semantics present in the first elements of our pairs. By applying the canonical rotation to the second of the two the idea is that we have isolated the key differences between the elements in a fixed frame of reference.

**Prototype Learning.** Canonicalize all semantic changes into the reference frame and average all the tangent vectors to calculate one Prototype $\vec{p}$, where $M$ is the total amount of sentence pairs[2]. This is a similar technique to mean-centering (Jorgensen et al., 2024):

$$\vec{p} = \frac{1}{M} \sum_{i=1}^{M} R(n_i) \, \log_{n_i}(v_i). \tag{2}$$

---

[2]For small angular differences, first-order equivalent to simply averaging the points and re-normalizing after the fact.

**Prediction.** To predict the semantic transformation for an unseen neutral embedding $n^*$, the prototype $\vec{p}$ can be used to predict the transformation embedding $v^*$ by converting the prototype $\vec{p}$ with the Riemannian exponential map and an application of the transpose of $n^*$'s canonicalizing rotor:

$$v^* = \exp_{n^*}(R(n^*)\top\vec{p}).  \tag{3}$$

$R(n^*)\top\vec{p}$ rotates $\vec{p}$ into the tangent space at $n^*$ . Then the Riemannian exponential $\exp_{n^*}(\vec{p})$ takes the tangent vector $\vec{p}$ and moves along the geodesic starting at $n^*$. The vector direction is which geodesic to follow and the length is how far along that arc to go (in radians).

## 4.2 DIFFERENTIATION FROM RELATED WORK

Our approach is related to recent advances in understanding linear representations in language models. As discussed in Section 2.2, Park et al. (2025) use a "causal inner product" that respects semantic structure in a function space using the Riesz isomorphism. However, RISE uses Riemannian geometry to operate consistently on the curved manifolds. Both methods take advantage of geometric properties, but the methods are distinctly different.

Crucially, RISE transformations exhibit commutativity: applying multiple semantic transformations yields consistent results regardless of order (see Appendix A). This commutativity property provides strong evidence for the LRH, as it demonstrates that semantic transformations behave like vector additions in the tangent space—geodesics serve as the curved-space generalization of straight lines. The preservation of additive structure across semantic operations suggests that the geometric framework captures fundamental algebraic properties of meaning composition. We discuss more about the commutativity properties in Appendix A.

Furthermore, the analysis in Park et al. (2025) focused on categorical relationships in the unembedding space of language models; our work examines discourse-level transformations in sentence embeddings across multiple languages. RISE effectively implements a non-Euclidean transformation that aligns with the natural curved manifold structure of the embedding space. This connection to high-dimensional geometry provides theoretical grounding for why rotational operations can capture semantic transformations more effectively than simple vector additions, and extends the linear subspace hypothesis to curved/geodesic subspaces.

## 5 EXPERIMENTAL DESIGN

### 5.1 DISCOURSE-LEVEL SEMANTIC-SYNTACTIC CHANGES & LANGUAGE SELECTION

We focus on three discourse-level semantic-syntactic transformations that vary in their context-dependence:

**Negation:** The logical reversal of the propositional content of a statement; where the proposition is "P" we take the negation to be "not-P." Moreso, we are negating the predicate. This transformation is semantically precise and should exhibit high geometric consistency across contexts and languages.

**Conditionality:** Converting declarative statements into conditional constructions ("P" → "If P"). This introduces modal semantics that may interact with contextual factors.

**Politeness:** Increasing the social formality or deference level of utterances. This is highly context-dependent and culturally variable, making it a challenging test case for geometric consistency.

We selected seven morphologically diverse languages to ensure broad coverage of morphological, syntactic phenomena, and resource levels: English, Spanish, Japanese, Tamil, Thai, Arabic, and Zulu. This selection spans multiple language families (Indo-European, Sino-Tibetan, Dravidian, Afroasiatic, Niger-Congo) and different morphological types (analytic, agglutinative, fusional). The languages also represent different levels of language model availability and resources. The diversity is crucial because different languages realize semantic transformations through distinct linguistic mechanisms. For instance, negation might be expressed through: (1) Particles (i.e. English "not"); (2) Affixes (i.e. Tamil verb-internal negation, Japanese "nai"); and (3) Auxiliary constructions (i.e. English "does/has not"). By testing across this range, we can determine whether geometric consistency reflects universal semantic properties or is merely an artifact of particular linguistic structures.

## 5.2 DATASETS, EMBEDDING MODELS, & LINEAR BASELINES

We use three datasets and three models for evaluation. We used two open-source, external datasets: **The Benchmark of Linguistic Minimal Pairs (BLiMP)** (Warstadt et al., 2020) and **Sentences Involving Compositional Knowledge (SICK)** (Marelli et al., 2014), and synthetically generated one dataset, referred to as the **Synthetic Multilingual** dataset. For each language-transformation combination in the Synthetic Multilingual dataset, we generated 1,000 neutral-transformed sentence pairs using GPT-4.5 with carefully controlled prompts (see Appendix D). To ensure robust analysis, we implemented several diversity controls (see Appendix E).

We compare three multilingual embedding models: **text-embedding-3-large** (OpenAI, 2024), **bge-m3** [3] (Chen et al., 2024), and **mBERT** (Devlin et al., 2019). The text-embedding-3-large model produces 3072-dimensional vectors, bge-m3 produces 1024-dimensional vectors, and mBERT produces 768-dimensional vectors. All selected models produce constant-length embeddings that reside on a hypersphere making them suitable for our geometric analysis. This dimensional diversity allows us to test whether RISE effectiveness depends on embedding dimensionality. We calculate a *rotor alignment score* where the scores represent mean cosine similarity between predicted embedding vectors and the semantically transformed pair on held-out test sets, with higher values indicating more consistent geometric structure. Table 1 describes how the cosine similarity scores are interpreted.

We include Mean Difference Vectors (MDV), and Procrustes alignment as baseline comparisons because they represent standard linear approaches used to model transformations in embedding spaces. MDV test whether simple difference vectors can capture semantic or cross-lingual structure, while Procrustes evaluates whether a single global rotation can align transformed embeddings. MDV is the geometrically correct analogue of the Euclidean additive method for modern spherical embeddings, providing a stronger and fairer baseline for RISE.

| Cosine Similarity Range | Interpretation | Supporting Literature |
|---|---|---|
| $\geq 0.80$ | Strong, consistent geometric structure | Reimers & Gurevych (2019) |
| 0.65–0.80 | Moderate, reliable structure | Mikolov et al. (2013b); Ethayarajh (2019) |
| 0.50–0.65 | Weak or variable structure | Ethayarajh (2019); Conneau et al. (2018) |
| $< 0.30$ | Inconsistent or failing transformation | Artetxe et al. (2018); Conneau et al. (2018) |

Table 1: Interpretation of cosine similarity magnitudes used throughout this work. Higher values indicate stronger geometric consistency between predicted and target embeddings. These thresholds are stricter than prior work but remain consistent with the established interpretations in the literature.

## 6 RESULTS

### 6.1 CROSS-LANGUAGE TRANSFER COMPARISON

This section discusses the comparison of embedding models trained in one of the seven languages and tested on all seven. The results of this section demonstrate RISE multilingual performance computed by three embedding models. See Appendix B for comprehensive results across all phenomena for each model.

**Negation** emerges as the most robust discourse-level, semantic-syntactic transformation, achieving the highest mean rotor alignment score **(0.788)** across all model-language combinations with performance ranging from 0.686 to 0.918. Figure 2 demonstrates RISE performance on negation for each model. RISE transformations for negation are most geometrically consistent in text-embedding-3-large. Negation's strong performance indicates that generalizable discourse-level, semantic-syntactic changes are captured by RISE and best applied cross-lingually in text-embedding-3-large.

---

[3]Bge-m3 should be m3 as titled in the final version of (Chen et al., 2024), but we referenced the model as bge-m3 in this paper and figures.

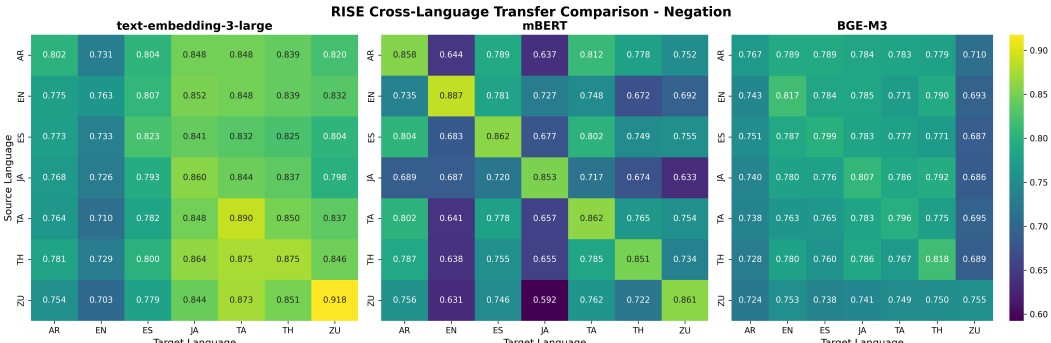

Figure 2: Embedding model heatmap cross-lingual transfer comparison on negation.

**Conditionality** demonstrates the highest stability and consistency across cross-language transfers, with the lowest performance variability (0.038) and most stable individual measurements (see Appendix B). With the second highest, mean performance **(0.780)**, conditionality is particularly consistent results across all combinations. The strong transfer seen in bge-m3 and text-embedding-3-large suggests that conditional semantics are captured by stable geometric structure despite their modal complexity.

**Politeness** exhibits the most variable geometric structure, ranking third in performance **(0.762 mean)** with the highest performance variability (0.060) across combinations. This variability aligns with expectations, as politeness realizations depend heavily on cultural context and linguistic conventions, making cross-language transfer inherently more challenging.

The contrast across phenomena performance reflects an interesting insight. In the results, negation appears more robust, politeness is most variable, and conditionality sits between. This suggests embeddings encode logical semantic operators (negation and conditionality) with strong cross-lingual consistency. However, pragmatic operators (politeness) are less reliable due to inherent language-specific indicators and cultural conventions. Additionally, cross-language analysis revealed dimensionality does not directly predict cross-lingual performance. Despite having lower dimensionality, bge-m3 (1024-dim) demonstrated the least variance in cross-language performance for all phenomena and languages. While text-embedding-3-large (3072-dim) showed highest cross-language performance (Figure 3), mBERT (768-dim) showed strong monolingual performance, but exhibited high variability, particularly for politeness in cross-language settings. These results highlight that training methodology and architectural choices matter more than raw embedding dimensionality for cross-language semantic transfer.

The cross-language analysis fully presented in Appendix B supports our hypothesis that discourse-level semantic-syntactic transformations correspond to geometric operations on the embedding manifold. The variation across models, preservation of linguistic relationships across languages, and transformation patterns indicate that RISE successfully identifies semantic-syntactic transformation on the embedding manifold. The limitations and future work are discussed further on.

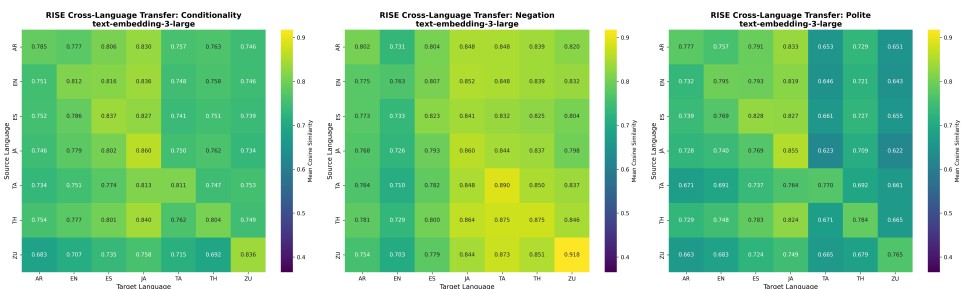

Figure 3: Cross-language transfer heatmaps for text-embedding-3-large showing RISE performance across all language pairs for conditionality, negation, and politeness transformations. Darker colors indicate higher cosine similarity between predicted and target embeddings.

## 6.2 CROSS-MODEL TRANSFER COMPARISON

To evaluate RISE prototypes' robustness to transfer across different embedding architectures, we conducted cross-model mapping experiments using the method developed by Morris et al. (2020). This approach learns statistical mappings between embedding spaces through principal component analysis (PCA) and distributional alignment, enabling transfer of learned RISE prototypes from one model to another. We specifically examined transfer from text-embedding-3-large (3072-dimensional) to bge-m3 (1024-dimensional), demonstrating cross-model semantic transfer across different dimensionalities and training objectives. For each language pair and phenomenon, we learn RISE prototypes in text-embedding-3-large using 80% of the data, map these prototypes and $e_1$ to bge-m3 space, and evaluate performance on native bge-m3 embeddings using the remaining 20%. Figure 4 demonstrates comprehensive cross-model and cross-language transfer results.

Cross-model transfer from text-embedding-3-large to bge-m3 reveals strong language-dependent performance. English achieves 0.80-0.82 similarity across all transformations, while other languages cluster around 0.70-0.75, and Zulu consistently scores 0.63-0.66. This 20% performance gap persists across conditionality, negation, and politeness transformations. These results suggest rotations can transfer between architecturally different models, but their effectiveness depends critically on source language, indicating that learned transformations are not architecture-independent. The consistent English advantage across models suggests these embedding spaces share more robust geometric structures for English, likely reflecting training data imbalances (Anglo-centric bias in the composition of the model's training data). The consistent language ranking across different semantic transformations (conditionality, negation, politeness) suggests the bias is structural rather than semantic. In conclusion, RISE successfully captures semantic patterns that perform consistently in a cross-model comparison.

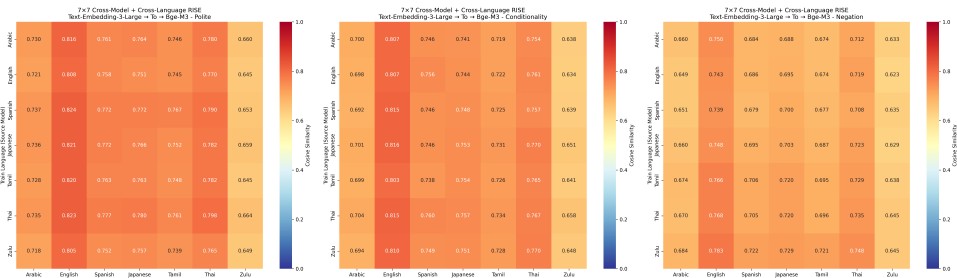

Figure 4: Cross-Model Semantic Transfer: text-embedding-3-large → bge-m3. Each cell shows transfer performance from source language prototype (text-embedding-3-large) to target language test set (bge-m3). Diagonal elements represent pure cross-model transfer, while off-diagonal elements show combined cross-model and cross-language transfer using Morris statistical mapping (Morris et al., 2020).

## 6.3 ENGLISH TASK-BASED COMPARISON

Our main investigation is how well RISE peforms in in multi-lingual settings. However there are limited external datasets for evaluating the performance discourse-level, semantic-syntactic transformation tasks. Due to the limited resources, we had to select the most related datasets, BLiMP and SICK. BLiMP is a paired sentence dataset for major grammatical phenomena in English, and SICK is a dataset with paired sentences with entailment, contradiction, and neutral labels.

Table 2 summarizes RISE performance across the three selected datasets. The results confirm that all models achieve strong performance, with particular strengths varying by dataset: mBERT excels on grammatical tasks (BLiMP) and contradiction detection (SICK), while bge-m3 shows the most consistent performance across synthetic multilingual data. The dramatic performance gap between BLiMP ($>0.92$) and SICK (0.62-0.74) suggests that RISE rotations might be capturing something more specific than general semantic transformations.

The high BLiMP performance indicates RISE excels at preserving grammatical/syntactic structure, while the moderate SICK performance suggests these same rotations don't preserve semantic re-

latedness as well. These results show that benchmark choice dramatically affects relative model ranking. Robustness depends on whether the task prioritizes cross-lingual consistency (favoring bge-m3) or raw performance on specific phenomena (favoring text-embedding-3-large for negation, mBERT for grammatical tasks).

Table 2: RISE Performance Across Three Datasets: The performance is measured with the rotor alignment score between RISE-steered embeddings and target embeddings where bold values indicate best performance per dataset. text-embedding-3-large is abbreviated as TE3L.

| Model | Synthetic Multilingual | BLiMP Benchmark | SICK Dataset |
|---|---|---|---|
| TE3L (3072d) | 0.771 | 0.929 | 0.623 |
| bge-m3 (1024d) | **0.782** | 0.956 | 0.631 |
| mBERT (768d) | 0.709 | **0.961** | **0.736** |
| **Average** | 0.754 | 0.949 | 0.663 |

## 6.4 LINEAR BASELINE COMPARISONS

The full results presented in Appendix C compare RISE against two standard baselines, Mean Difference Vectors (MDV) and Procrustes alignment, across the same three datasets. MDV is not Euclidean. MDV preserves spherical structure and naturally resembles RISE more closely than Procrustes. This distinction is directly reflected in the results: MDV and RISE transfers best across languages where Procrustes fails.

The strongest performance appears in monolingual English evaluation (BLiMP), while performance drops substantially for Procrustes on semantic relatedness (SICK) shown in Table 3. This shift in performance reflects Procrustes' inability to identify a generalizable semantic–syntactic relationship as expected by method. Procrustes fits a single global rotation which is too rigid for the cross-lingual and cross model analysis In contrast, RISE maintains stable cross-lingual and cross-model performance (e.g., App. B. Figures 5–7), indicating that geometric operations on the manifold better capture discourse-level semantic structure than Euclidean differences.

The MDV vs. RISE vs. Procrustes results reinforce our earlier claim that methods operating on the curved manifold (where sentence embeddings inherently reside) perform better than Euclidean/linear methods. Most steering and probing techniques operate in linear space, and we conjecture that this geometric mismatch helps explain why linear methods struggle to generalize. In short, Procrustes fits a single global rotation which is too rigid for the cross-lingual and cross model analysis. Geometric transformations, like RISE and MDV, are better suited for semantic-syntactic analysis and cross-lingual stability.

| Method | Monolingual Syntactic (BLiMP) | Monolingual Semantic (SICK) | Cross-Language Transfer (All Phenomena) |
|---|---|---|---|
| **RISE** | **Strong** (0.97) | **Strong** (0.84) | **Moderate–Strong** (0.74–0.89) |
| **MDV** | **Strong** (0.97) | **Strong** (0.83) | **Moderate–Strong** (0.72–0.91) |
| **Procrustes** | **Strong** (0.99) | **Moderate** (0.67) | **Failing–Weak** (0.25–0.62) |

Table 3: Condensed summary of baseline comparisons from Appendix C using the cosine-similarity interpretation scale from Table 1. RISE and MDV show Strong monolingual and Moderate–Strong cross-language structure, whereas Procrustes drops to Weak or Failing consistency outside syntactic, same-language settings.

## 7 DISCUSSION & FUTURE WORK

Our findings demonstrate that meaningful semantic-syntactic operations can be recovered as geometric transformations in modern language model representations. RISE successfully identifies consistent geometric structure for discourse-level semantic-syntactic changes, primarily for text-embedding-3-large and negation in multilingual settings. The results demonstrating spherical meth-

ods, RISE and MDV, out perform linear methods, Procrustes alignment, provide positive results for extending the LRH to spherical spaces.

Evaluation benchmarks (Table 2) reveal task-dependent effectiveness. RISE achieves near-perfect performance on syntactic acceptability (BLiMP: 0.93-0.96) but only moderate performance on semantic similarity (SICK: 0.62-0.74), suggesting better alignment with grammatical rather than semantic transformations. Section 6.1 shows that negation and conditionality are the most generalizable discourse-level, semantic-syntactic changes captured by RISE and best applied cross-lingually in text-embedding-3-large. Our cross-model transfer experiments expose an English-centric bias, with English achieving 20% higher transfer scores than languages like Zulu. This English-centric bias persists across all semantic transformations, indicating that current multilingual models encode geometric structures that prioritize English. Future work should focus on developing more equitable multilingual representations and investigating which language-specific geometric structures are an inherent feature of the models.

Together these results support that RISE is most successful at identifying semantic transformation with distinct grammatical factors, but more work is needed to justify semantic transformations in multilingual models are universal geometric operations. First, our analysis focuses on three specific linguistic transformation types. Future work should expand to additional semantic and pragmatic phenomena to test the generality of geometric consistency principles. Second, while our experiments used three diverse embedding models (text-embedding-3-large, bge-m3, and mBERT), validation across additional architectures would strengthen claims about the universality of geometric semantic structure. Third, the reliance on GPT-4.5 for data generation may introduce subtle biases toward English-centric conceptualizations of semantic phenomena. Future work should incorporate more diverse data sources and validation by native speakers.

## 8    CONCLUSION

The ability to learn geometric transformations for discourse changes relates to work on text generation and steering vectors (Turner et al., 2023; Li et al., 2023). Our rotor-based approach, RISE, provides a geometric framework for understanding and improving interpretability in language models. This work investigated whether discourse-level semantic-syntactic transformations in multilingual embedding spaces correspond to intrinsic geometric operations, specifically rotations identified through the RISE method. Our comprehensive evaluation across multiple baselines, models, languages, and datasets reveals a more complex reality than initially hypothesized. This work demonstrates that modern language model representations maintain interpretable geometric structure for some semantic-syntactic transformations, extending the promise of geometric semantics from early word embeddings to contemporary transformer models. We show that:

1. Semantic transformations with clear syntactic mapping demonstrate the most consistent geometric structure.

2. RISE successfully identifies semantically meaningful geometric structure in high-dimensional embedding spaces that generalizes cross-lingually and across model architecture.

As language models continue to evolve, understanding these geometric foundations will be crucial for developing more interpretable AI systems. By revealing transferable geometric structure in semantic transformations (e.g. negation and conditionality), this work opens new possibilities for understanding language model behavior through geometric interventions. Our work promotes geometric methods as more appropriate approaches to cross-lingual semantic interpretation, achieving 77%-95% cross-language transfer effectiveness across typologically diverse languages. By developing RISE, we demonstrate that interpretable structure exists for some grammatically distinct semantic transformations, providing a tools for understanding how these systems encode semantic knowledge. While RISE remains valuable for analyzing model-specific semantic structures, claims about universal geometric operations require substantial qualification.

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

## A  MATHEMATICAL PROPERTIES OF RISE

These mathematical results support our main claims in the paper. Lemma 1 provides the explicit exponential and logarithmic map formulas that underlie RISE's use of geodesics on the unit hypersphere. Theorem A.1 formalizes that sequential RISE edits commute up to second order, showing that different discourse-level transformations can be applied in any order without significant distortion. This result highlights the local geometric consistency of RISE transformations, rather than implying global additive steering. Proposition A.1 shows that each RISE transformation can be applied in $O(d)$ time and memory, demonstrating the method's scalability to modern high-dimensional embeddings. Together, these results provide theoretical grounding for both the geometric consistency and the practical efficiency reported in the main text.

### A.1  GEOMETRY PRELIMINARIES ON THE SPHERE

We work on the unit sphere $\mathbb{S}^{d-1} \subset \mathbb{R}^d$ with the standard round metric. For $n \in \mathbb{S}^{d-1}$, the tangent space is $T_n\mathbb{S}^{d-1} = \{x \in \mathbb{R}^d : \langle x, n \rangle = 0\}$. The exponential map $\exp_n : T_n\mathbb{S}^{d-1} \to \mathbb{S}^{d-1}$ is defined for all tangent vectors, while the logarithmic map $\log_n$ is well-defined for all $v \in \mathbb{S}^{d-1}$ except the antipode $v = -n$. For each $n$, fix an orthogonal map $R(n) \in O(d)$ such that $R(n)n = e_1$, where $e_1 = (1, 0, \ldots, 0)^\top$. When analyzing local behavior (e.g., Theorem A.1), we take $R(\cdot)$ to be any $C^1$ (continuously differentiable) choice on a neighborhood of the geodesic segment(s) under consideration; such a local choice always exists.

**Lemma 1** (Exponential and logarithmic maps on the unit sphere). *For $n \in \mathbb{S}^{d-1}$, tangent vector $\xi \in T_n\mathbb{S}^{d-1}$, and point $v \in \mathbb{S}^{d-1} \setminus \{-n\}$,*

$$\exp_n(\xi) = \cos(\|\xi\|)\, n + \sin(\|\xi\|)\, \frac{\xi}{\|\xi\|}, \qquad \log_n(v) = \arccos(\langle n, v \rangle)\, \frac{v - \langle n, v \rangle n}{\|v - \langle n, v \rangle n\|}.$$

*Proof.* These formulas follow from the fact that geodesics on $\mathbb{S}^{d-1}$ are great circles in $\mathbb{R}^d$ (unit-radius sphere). See, e.g., Absil et al. (2008, Sec. 5.4).  □

A.2 ROTOR CONSTRUCTION AND IMPLEMENTATION

In Clifford algebra terms, a *rotor* is an element of $\mathrm{Spin}(d)$ that rotates vectors by the sandwich product $x \mapsto r x \tilde{r}$, where $\tilde{r}$ denotes reversion. For our purposes, we only require an orthogonal operator $R(n) \in O(d)$ with $R(n)n = e_1$ that depends smoothly on $n$. One closed-form rotor mapping $n \mapsto e_1$ (valid when $n \neq -e_1$) is

$$r(n) = \frac{1 + e_1 n}{\sqrt{2(1 + \langle e_1, n \rangle)}}, \qquad r(n)\, n\, \tilde{r}(n) = e_1.$$

In practice we realize this as a standard linear operator without explicit Clifford algebra structures. Two efficient $O(d)$ realizations are:

- **Householder reflection:** $H(n) = I - 2\frac{ww^\top}{\|w\|^2}$ with $w = n - e_1$, which satisfies $H(n)n = e_1$ (determinant $-1$).
- **Givens rotation:** a $2 \times 2$ rotation in the plane spanned by $\{n, e_1\}$, extended by the identity elsewhere, with determinant $+1$.

Both satisfy the required conditions $R(n)n = e_1$ and local $C^1$ smoothness, and are numerically stable away from $n \approx -e_1$. In the antipodal case ($n \approx -e_1$) we use a two-step construction: map $n$ to an auxiliary orthogonal vector $u \perp e_1$, then $u$ to $e_1$. In all cases, applying $R(n)$ or $R(n)^\top$ to a vector costs $O(d)$ operations.

A.3 COMMUTATIVITY PROPERTIES OF SEQUENTIAL RISE OPERATIONS

A.3.1 THE RISE SEQUENTIAL PROCEDURE

Given $n_0 \in \mathbb{S}^{d-1}$ and prototypes $\vec{p}_A, \vec{p}_B \in T_{e_1}\mathbb{S}^{d-1}$:

**Apply A:** $\xi_A = R(n_0)^\top \vec{p}_A,\ n_1 = \exp_{n_0}(\xi_A),$ **Apply B:** $\xi_B = R(n_1)^\top \vec{p}_B,\ n_2 = \exp_{n_1}(\xi_B).$

A.3.2 FIRST-ORDER COMMUTATIVITY ANALYSIS

**Theorem A.1** (RISE commutativity to first order). *For small prototype magnitudes* $\|\vec{p}_A\|, \|\vec{p}_B\| \ll 1$,

$$d(\text{result of } A \circ B,\ \text{result of } B \circ A) = O(\|\vec{p}_A\| \cdot \|\vec{p}_B\|).$$

*Proof.* Using Lemma 1, expand $\exp_{n_0}(\xi_A) = n_0 + \xi_A + O(\|\xi_A\|^2)$. Let $\eta_A = \xi_A$. Canonicalization at $n_1 = n_0 + \eta_A + O(\|\eta_A\|^2)$ differs from that at $n_0$ by $O(\|\eta_A\|)$.

Let $P_{n_1 \to n_0} : T_{n_1}\mathbb{S}^{d-1} \to T_{n_0}\mathbb{S}^{d-1}$ denote parallel transport along the short geodesic from $n_1$ to $n_0$. On the unit sphere, $\|P_{n_1 \to n_0} - I\| = O(\|n_1 - n_0\|) = O(\|\eta_A\|)$, where $I$ denotes the identity operator on the tangent space. With a $C^1$ choice of $R(\cdot)$, $\|R(n_1)^\top - R(n_0)^\top\| = O(\|n_1 - n_0\|) = O(\|\eta_A\|)$. Therefore,

$$P_{n_1 \to n_0}\, R(n_1)^\top \vec{p}_B = R(n_0)^\top \vec{p}_B + O(\|\eta_A\|\,\|\vec{p}_B\|).$$

Now expand the second step:

$$n_2 = n_0 + R(n_0)^\top(\vec{p}_A + \vec{p}_B) + O(\|\vec{p}_A\|\|\vec{p}_B\|) + O(\|\vec{p}_A\|^2 + \|\vec{p}_B\|^2).$$

Swapping roles of $A$ and $B$ gives the same expansion with $\vec{p}_A, \vec{p}_B$ reversed. Subtracting yields a difference of order $\|\vec{p}_A\|\|\vec{p}_B\|$. $\square$

**Geometric interpretation.** Re-canonicalization is equivalent (to first order) to parallel-transporting the next step's vector back to the initial tangent space. On $\mathbb{S}^{d-1}$ with constant curvature, order effects are second order.

A.4 COMPUTATIONAL COMPLEXITY

**Proposition A.1** (Per-transformation complexity). *Each RISE transformation can be implemented in $O(d)$ time and $O(d)$ memory:*

1. *Canonicalization: applying $R(n)$ or $R(n)^\top$ costs $O(d)$.*

2. *Logarithmic map $\log_n(v)$: $O(d)$ using Lemma 1.*

3. *Exponential map $\exp_n(\xi)$: $O(d)$ using Lemma 1.*

4. *Storage: prototype $\hat{\vec{p}} \in T_{e_1}\mathbb{S}^{d-1}$ costs $O(d)$.*

**Comparison with matrix methods.** Dense $d \times d$ rotations require $O(d^2)$ time and memory. RISE achieves equivalent updates in $O(d)$.

**Implementation note (Householder).** A practical canonicalization is the Householder reflection

$$H(n) = I - 2\frac{ww^\top}{\|w\|^2}, \quad w = n - e_1,$$

which maps $n \mapsto e_1$ in $O(d)$. Since $H(n)$ is a reflection ($\det = -1$), it suffices for canonicalization. Near $n \approx e_1$, one may switch to a numerically stable alternative.

# B  CROSS-LANGUAGE TRANSFER ANALYSIS AND RESULTS

To test whether geometric transformations generalize across languages, we conducted comprehensive cross-language transfer experiments. This section reports detailed results across three models and three semantic phenomena, analyzing both quantitative performance and geometric properties of learned transformations.

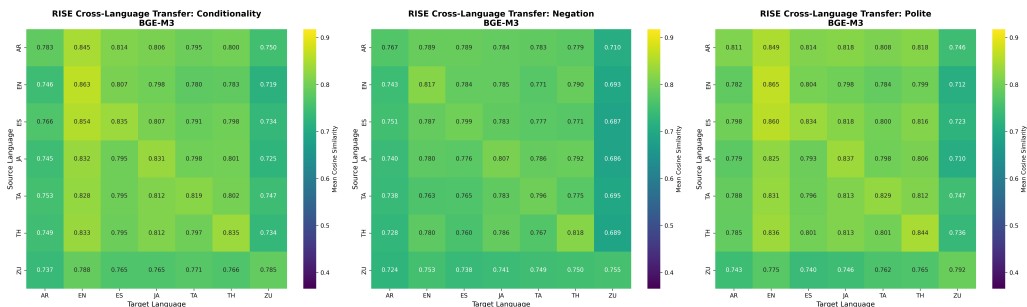

Figure 5: Cross-language transfer heatmaps for bge-m3 model showing RISE performance across all language pairs for conditionality, negation, and politeness transformations. Darker colors indicate higher cosine similarity between predicted and target embeddings.

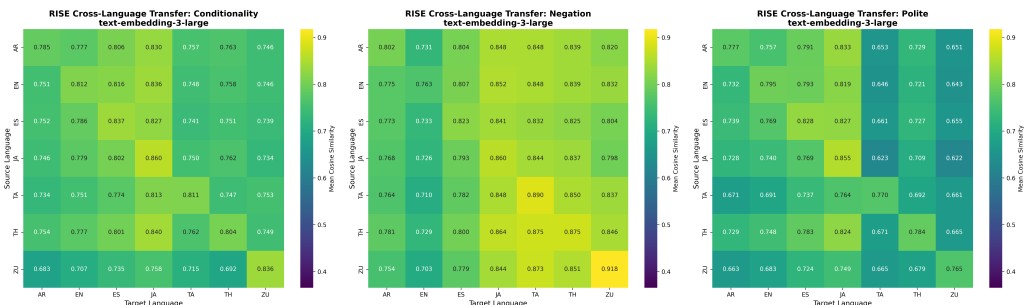

Figure 6: Cross-language transfer heatmaps for text-embedding-3-large model showing RISE performance across all language pairs for conditionality, negation, and politeness transformations. Darker colors indicate higher cosine similarity between predicted and target embeddings.

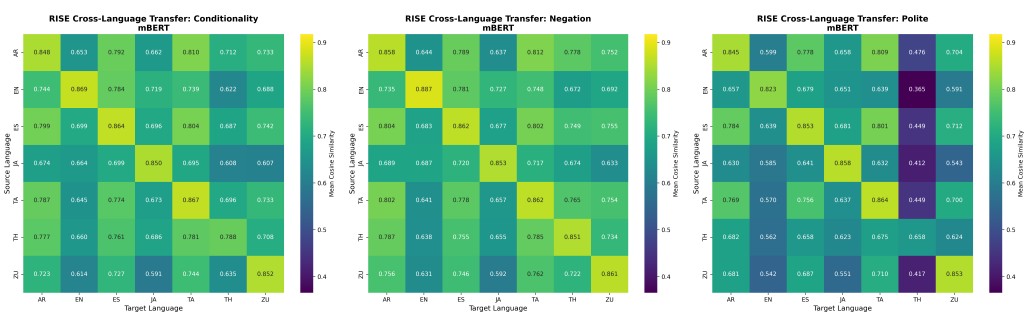

Figure 7: Cross-language transfer heatmaps for mBERT model showing RISE performance across all language pairs for conditionality, negation, and politeness transformations. Darker colors indicate higher cosine similarity between predicted and target embeddings.

## B.1 Cross-Language Transfer Performance

The above heatmaps demonstrate comprehensive cross-language transfer results across our three models. Training rotor prototypes on one language and evaluating on others reveals promising cross-linguistic performance, particularly for negation and conditionality. Most language pairs show transfer scores above 0.70, with negation achieving particularly strong off-diagonal performance (most scores > 0.80).

**Negation** emerges as the most performant transformation, achieving the highest mean cross-language transfer scores (0.788 across all model-language combinations) with performance ranging from 0.686 to 0.918.

**Conditionality** demonstrates the highest stability and consistency across cross-language transfers, with the lowest performance variability (0.038) and most stable individual measurements (0.056 average std deviation). Mean performance of 0.780 places it second overall.

**Politeness** shows more variation but still achieves substantial cross-linguistic success (most scores > 0.70).

## B.2 Geometric Analysis of Cross-Language Centroids

Analysis of the learned centroids reveals additional insights into the geometric structure of semantic transformations. For each phenomenon, we computed "ideal" transformation vectors by averaging canonicalized transformed embeddings across languages.

For **negation**, the centroids show high similarity across languages (pairwise cosines > 0.95).

**Conditionality** centroids maintain high geometric consistency, supporting the observed stability in transfer performance across all model-language combinations.

**Politeness** centroids cluster more loosely but still maintain substantial similarity (pairwise cosines > 0.87).

### B.3 QUANTITATIVE CROSS-LANGUAGE ANALYSIS

Table 4: Complete Cross-Language Transfer Matrix: Statistical Summary

| Model | Phenomenon | All Transfers | Monolingual | Cross-Lang | Ratio |
|---|---|---|---|---|---|
| | Conditionality | 13.6× ± 0.7 | 14.5× | 13.5× | 0.93 |
| TE3L (3072d) | Negation | 19.7× ± 1.2 | 20.6× | 19.6× | 0.95 |
| | Politeness | 23.1× ± 1.9 | 25.3× | 22.8× | 0.90 |
| | Conditionality | 13.9× ± 0.6 | 14.5× | 13.8× | 0.95 |
| bge-m3 (1024d) | Negation | 18.5× ± 0.8 | 19.3× | 18.4× | 0.95 |
| | Politeness | 25.2× ± 1.2 | 26.4× | 25.1× | 0.95 |
| | Conditionality | 12.8× ± 1.3 | 15.0× | 12.5× | 0.83 |
| mBERT (768d) | Negation | 18.0× ± 1.8 | 20.9× | 17.5× | 0.84 |
| | Politeness | 20.8× ± 3.9 | 26.1× | 20.0× | 0.77 |

Statistics computed across complete 7×7 language transfer matrix (49 language pairs per phenomenon).
Values show advantage ratios ± standard deviation across all language pairs.
Ratio indicates relative cross-language transfer effectiveness (Cross-Lang/Monolingual).
All models maintain strong cross-language performance (77%–95% of monolingual performance).

Table 5: Model Architecture and Overall RISE Performance Summary

| Model | Dims | Validation Avg | Cross-Lang Avg | Random Adv |
|---|---|---|---|---|
| TE3L | 3072 | 0.774 | 19.0× | 6.3× |
| bge-m3 | 1024 | **0.790** | **19.8×** | 11.7× |
| mBERT | 768 | **0.802** | 16.9× | **11.9×** |

Validation Avg: Mean performance across Synthetic Multilingual, BLiMP, and SICK datasets.
Cross-Lang Avg: Mean advantage ratio across English→Spanish and Japanese→English transfers.
Random Adv: Mean advantage ratio over random baselines in monolingual English scenarios.
Bold values indicate best performance in each category.

Tables 4 and 5 provide comprehensive quantitative analysis of cross-language transfer performance. Notably, all models maintain strong cross-language performance (77%–95% of monolingual performance), with bge-m3 showing the most consistent cross-language effectiveness across all phenomena.

## C    LINEAR BASELINES COMPARISONS

This appendix reports the full results for the linear baseline comparisons requested by the reviewers. We thank the reviewers for this valuable suggestion as these results did strengthen our paper. We implemented two baselines: Procrustes alignment and Mean Difference Vectors (MDV). MDV is not truly Euclidean: it computes mean displacements using the manifold's geometry (via log/exp maps), preserving spherical structure. Thus MDV functions naturally resembles RISE more closely than Procrustes. We evaluated them alongside RISE on three datasets: BLiMP, SICK, and our multilingual synthetic dataset.

The strongest performance appears in monolingual English evaluation (BLiMP), while performance drops substantially for Procrustes on semantic relatedness (SICK) shown in Table 3. This shift in performance reflects Procrustes' inability to identify a generalizable semantic–syntactic relationship as expected by method. Procrustes fits a single global rotation which is too rigid for the cross-lingual and cross model analysis In contrast, RISE maintains stable cross-lingual and cross-model performance (e.g., App. B. Figures 5–7), indicating that geometric operations on the manifold better capture discourse-level semantic structure than Euclidean differences.

The MDV vs. RISE vs. Procrustes results reinforce our earlier claim that methods operating on the curved manifold (where sentence embeddings inherently reside) perform better than Euclidean/linear methods. Most steering and probing techniques operate in linear space, and we conjecture that this geometric mismatch helps explain why linear methods struggle to generalize. In short, Procrustes fits a single global rotation which is too rigid for the cross-lingual and cross model analysis. Geometric transformations, like RISE and MDV, are better suited for semantic-syntactic analysis and cross-lingual stability.

### C.1    CROSS-LANGUAGE TRANSFER HEATMAPS

Figures 8–10 show cross-language cosine similarity for the three semantic transformations (Conditionality, Negation, Politeness) under Mean Difference Vectors (MDV), Procrustes alignment, and RISE.

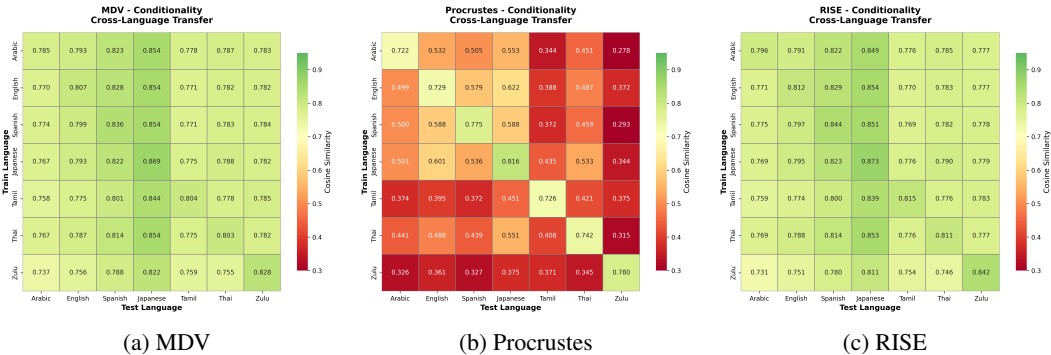

Figure 8: Cross-language transfer for **Conditionality** across seven languages.

### C.2    NATURAL-LANGUAGE VALIDATION: BLIMP AND SICK

Figure 11 reports mean cosine similarity on BLiMP (syntactic) and SICK (semantic) for the three methods.

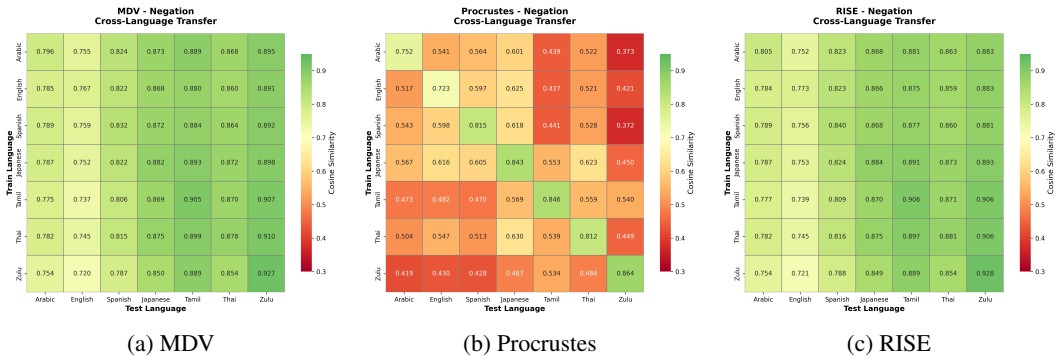

(a) MDV                              (b) Procrustes                              (c) RISE

Figure 9: Cross-language transfer for **Negation** across seven languages.

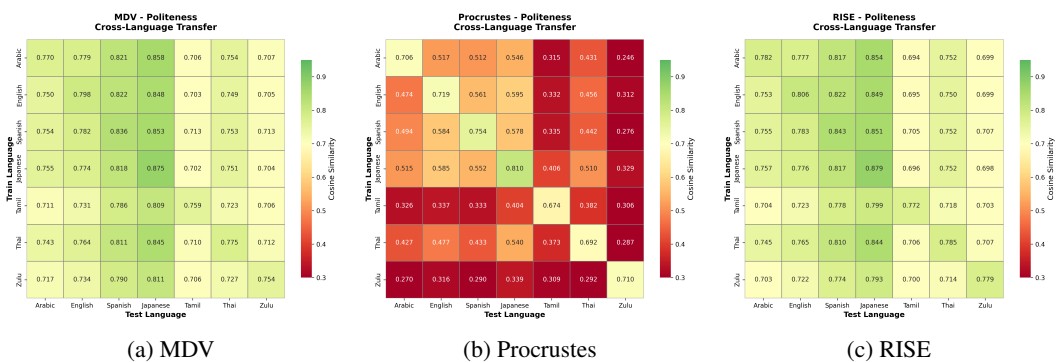

(a) MDV                              (b) Procrustes                              (c) RISE

Figure 10: Cross-language transfer for **Politeness** across seven languages.

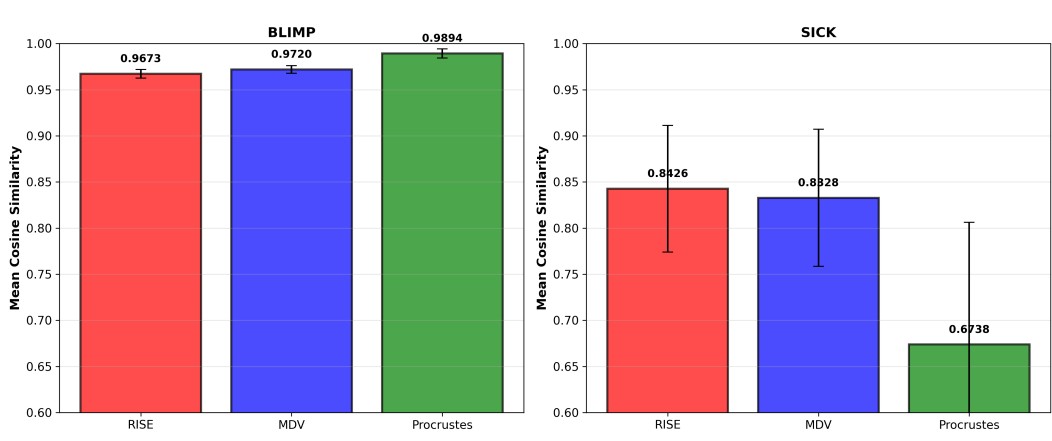

Figure 11: Natural language validation on BLiMP (syntactic acceptability) and SICK (semantic relatedness) for RISE, MDV, and Procrustes. Error bars denote standard deviation across examples.

## D PROMPT TEMPLATES

We provide the exact prompt templates used to generate neutral sentences and their semantic variants. Each template is shown in monospace using the `lstlisting` environment for clarity and reproducibility.

### D.1 NEUTRAL SENTENCE GENERATION

```
You are a linguistics assistant. Generate ONE terse, blunt English
sentence that is politeness-neutral: it must be neither explicitly
polite nor impolite. Keep it concise (8 to 12 words), direct, and
free of polite markers such as "please", honorifics, hedging,
or apologies, yet ensure it is not rude. If the situation contains
a placeholder (e.g., "a favor", "a cultural practice"), replace
it with a concrete, plausible example.

Context category: {category}
Detailed situation: {example}

Respond with ONLY the single sentence (no explanations, no quotation
    marks).
```

### D.2 POLITENESS REPHRASING

```
You are an expert translator and pragmatics specialist. Rewrite the
following sentence in {language_name} to make it more POLITE while
preserving its original meaning. Incorporate the given politeness
features.

Sentence: "{sentence}"

Politeness features (JSON): {features_json}

Respond ONLY with a JSON object in the exact format:
{"polite": "<rewritten sentence>"}
Do NOT add any other keys, explanations, or markdown.
```

### D.3 NEGATION

```
You are an expert translator and semantics specialist. Rewrite the
following sentence in {language_name} so that it expresses the
NEGATION of its original meaning while remaining natural and fluent.
Incorporate the given negation features.

Sentence: "{sentence}"

Negation features (JSON): {features_json}

Respond ONLY with a JSON object in the exact format:
{"negation": "<rewritten sentence>"}
Do NOT add any other keys, explanations, or markdown.
```

### D.4 CONDITIONALITY

```
You are an expert translator and syntax/pragmatics expert. Rewrite
the following sentence in {language_name} so that the statement
becomes CONDITIONAL (i.e., it only holds under a certain condition)
while preserving overall meaning and sounding natural. Incorporate
the provided conditionality features.

Sentence: "{sentence}"
```

```
Conditionality features (JSON): {features_json}

Respond ONLY with a JSON object in the exact format:
{"conditionality": "<rewritten sentence>"}
Do NOT add any other keys, explanations, or markdown.
```

# E   DATA GENERATION METHODOLOGY

## E.1   DIVERSITY CONTROLS

To guard against artifacts that might arise from narrow lexical or topical coverage, we apply several sampling diversity control. (1) Each neutral sentence prompt draws its situation description from a randomly chosen context category and exemplar, yielding a wide topical spread before any transformation is applied. (2) Within every language we shuffle sentence–feature assignments so that no specific lexical field correlates with a particular transformation subtype. (3) For each transformation we uniformly sample property values (e.g., negation particle, politeness strategy) per language and sentence, guaranteeing that every combination of language and subtype appears the same number of times. (4) After generation we remove near-duplicates and enforce a 5–25 token length window, which empirically yields a near-uniform length distribution. Together these steps ensure that our corpus varies in topic, syntax, and lexical choice while remaining balanced across languages and transformation subtypes. These controls ensure that observed geometric patterns reflect semantic properties rather than artifacts of lexical choice or sentence structure.

1. **Topical Diversity:** Neutral sentences were drawn from varied context categories (social interactions, factual statements, requests, etc.)

2. **Feature Balance:** Transformation features (e.g., negation particles, politeness strategies) were uniformly sampled to prevent correlation with specific lexical fields.

3. **Length Normalization:** Sentences were filtered to 5-25 tokens to ensure comparable embedding properties.

4. **Deduplication:** Near-duplicate outputs were removed to prevent repeated data.

## E.2   FEATURE-BASED TRANSFORMATION METHODOLOGY

We generated sentence pairs systematically by first sampling neutral sentences in seven typologically diverse languages (English, Spanish, Tamil, Thai, Arabic, Japanese, and Zulu), and subsequently transforming each sentence using feature-controlled prompts. Each transformation was guided by uniformly sampling linguistic features from a predefined typological metadata set (illustrated below).

The full inventories of typological properties for politeness, negation, and conditionality are provided in Tables 6–8.

### E.2.1   TRANSFORMATION PROCEDURE

For each neutral sentence, we uniformly sampled exactly one set of feature values from the typological metadata and prompted the language model (GPT-4.5) to generate the transformed variant adhering to these specifications. By uniformly sampling across multiple typological dimensions—strategy types, morphological realizations, and pragmatic contexts—we ensured comprehensive coverage of each language's linguistic variability. This methodology supports cross-linguistic embedding analysis and ensures that observed embedding-space transformations reflect typological distinctions accurately.

## E.3   FEATURE-CONTROLLED PROMPTING

To generate each transformation in a systematic and reproducible manner, we employ a feature-controlled prompting strategy with a large language model (LLM). Each prompt is carefully templated to specify the source language, the desired transformation type, and a set of fine-grained

| Language | Strategy Type | Grammatical/Lexical Devices |
|---|---|---|
| English | Negative politeness | Modal conditional, hedging, idiomatic/proverbial, taboo avoidance |
| Spanish | Positive politeness | Modal conditional, morphological politeness, hedging, idiomatic/proverbial |
| Tamil | Relational/Kinship politeness | Morphological politeness |
| Thai | Positive politeness; Relational/Kinship | Morphological politeness, modal conditional |
| Arabic | Positive politeness; Relational/Kinship | Modal conditional, morphological politeness, idiomatic/proverbial |
| Japanese | Relational/Kinship politeness | Morphological politeness, modal conditional, hedging |
| Zulu | Relational/Kinship politeness | Morphological politeness |

Table 6: Typological features sampled uniformly for politeness transformations.

| Language | Marker Position | Morphological Realization |
|---|---|---|
| English | Clause-medial | Negative particle; negative aux/modal; negative affix |
| Spanish | Clause-medial; concord | Negative particle |
| Tamil | Clause-final | Negative particle; verb-internal negation |
| Thai | Clause-medial | Negative particle |
| Arabic | Clause-initial / medial | Negative particle; negative affix |
| Japanese | Clause-final | Verb-internal negation |
| Zulu | Clause-medial | Negative particle |

Table 7: Typological features sampled uniformly for negation transformations.

feature tags that guide the model's output. For example, a prompt might indicate the language code ("[TA]" for Tamil), the transformation ("Politeness Rephrase"), and a particular strategy or keyword (such as "add honorific") relevant to that transformation. By explicitly encoding these features, we ensure that the LLM produces the intended variation—whether a more polite rephrasing, a negated statement, or a conditional construction—in a consistent and transparent way.

To further guarantee balanced coverage, we maintain a metadata table that enumerates all possible sub-types or strategies for each transformation. This enables us to stratify the sampling of transformation features across languages and sentences, ensuring that every variant type is equally represented. For instance, multiple politeness strategies (e.g., adding honorifics, using indirect language) or different negation words ("no" vs. "not") are distributed uniformly across the dataset. This controlled coverage is critical for fair comparisons: it prevents any language from being overrepresented by a particular style of rephrasing or negation, and minimizes inadvertent correlations between language and transformation realization. Our stratified sampling approach follows established principles of controlled experimental design, providing a robust foundation for cross-lingual embedding analysis.

All transformed sentences are generated using a single, consistent LLM (GPT-4.5) with a temperature of 1.0 and a maximum token limit of 128 per prompt. The relatively high temperature encourages diversity in phrasing, while the one-shot generation policy (taking the first model output without retries or manual curation) avoids selection bias. With carefully constructed prompts, the model reliably produces valid transformations on the first attempt, and all outputs remain in the target language specified by the prompt. This procedure ensures that our dataset is both systematically varied and reproducible, supporting rigorous downstream analysis.

| Language | Clause Structure | Morphological Marking |
|----------|------------------|-----------------------|
| English | Initial; final; embedded | Explicit marker; conditional tense/aspect |
| Spanish | Initial; final; embedded | Conditional mood; explicit marker |
| Tamil | Final | Explicit marker; conditional mood |
| Thai | Initial | Explicit marker |
| Arabic | Initial; final | Conditional mood; explicit marker |
| Japanese | Final; embedded | Conditional mood; explicit marker |
| Zulu | Initial | Conditional mood; explicit marker |

Table 8: Typological features sampled uniformly for conditionality transformations.

### E.4 QUALITY CONTROL AND DEDUPLICATION

To ensure the integrity and uniqueness of our dataset, we implemented a rigorous two-level deduplication process. At the first level, we removed any transformed sentence that was exactly identical to another within the same category and language. This step addresses the possibility that the LLM might produce identical outputs for different inputs, especially for short or formulaic sentences. At the second level, we ensured that each (neutral, variant) pair was unique across the entire dataset. In rare cases where two different source sentences yielded the same transformed output, we treated this as a collision and regenerated a new variant using a slightly altered prompt. Through this process, every neutral sentence in our dataset is paired one-to-one with three distinct transformed sentences (one per transformation type), with no overlaps. The result is a clean set of sentence pairs, each exhibiting a unique, transformation-driven difference.

Beyond deduplication, we applied a suite of diversity controls to guard against artifacts arising from narrow lexical or topical coverage. Each neutral sentence prompt was drawn from a wide range of context categories and exemplars, ensuring topical breadth before any transformation was applied. Within each language, we shuffled sentence–feature assignments so that no specific lexical field correlated with a particular transformation subtype. For each transformation, we uniformly sampled property values (such as negation particles or politeness strategies) per language and sentence, guaranteeing that every combination of language and subtype appeared the same number of times. After generation, we removed near-duplicates and enforced a 5–25 token length window, which empirically yielded a near-uniform length distribution. Together, these steps ensure that our corpus varies in topic, syntax, and lexical choice while remaining balanced across languages and transformation subtypes, providing a robust foundation for subsequent embedding analysis.

### E.5 EMBEDDING GENERATION

With our dataset of neutral and transformed sentences in hand, we next obtain high-dimensional vector representations using a state-of-the-art multilingual sentence encoder. Specifically, we employ OpenAI's `text-embedding-3-large` model, which produces 3072-dimensional embeddings aligned semantically across more than 90 languages.[4] All embeddings are generated in a frozen (non-fine-tuned) setting, with a single API call per sentence. According to the model card, each sentence embedding is computed by mean-pooling the token-level hidden states, followed by layer normalization. This means that every token—including short functional items like negation particles—contributes proportionally to the final vector.

Our approach assumes that all sentence embeddings reside in a shared semantic space where linear structure is meaningful. We adopt the perspective that this space forms a latent manifold encoding universal semantic features, as hypothesized by Jha et al. (2025). In this framework, certain directions in the embedding space correspond to specific attributes, such as politeness or negation. If sentence transformations truly correspond to adding or subtracting a semantic attribute, we expect

---

[4]https://platform.openai.com/docs/guides/embeddings

the difference vector (variant minus source) to be relatively consistent across examples. This aligns with the "universal geometry for embeddings" framework, in which multilingual embeddings from different models or languages can be brought to a common representation where semantic differences are captured by geometric translations. While our work stays within a single encoder's space, we leverage a similar idea: analyzing whether the transformation "rotors" (difference vectors) cluster for similar transformations across languages. This methodology sets the stage for validating whether these quasi-linear transformations indeed behave like translations in a Riemannian semantic space (Jha et al., 2025), which we explore in the next section via rotor-based analysis of the embedding differences.

It is important to note that applying a single global rotation or principal component analysis (PCA) can distort other dimensions and is not adaptive to individual vectors. Because the base embedding is already a mean across tokens, edits that insert or replace a handful of tokens translate to small but coherent rotations of the global vector—precisely the kind of local, content-independent shift that our rotor method is designed to capture.

### E.6 FINAL DATASET STATISTICS

The resulting corpus comprises 1,000 neutral sentences in each of the seven languages, totaling 7,000 examples. For English neutral sentences, the mean token length is 9.1 tokens (with a median of 9.0 tokens), with token counts ranging from three to 12 tokens and an average character length of 54.4 characters. This distribution confirms that our generation process produced concise, natural sentences suitable for semantic transformation analysis across languages and transformation types.

To further validate the diversity and balance of our dataset, we analyzed the distribution of sentence lengths per language, which reveals broadly similar profiles with a peak around 10–15 tokens. Additionally, we examined the distribution of word frequencies, confirming a typical long-tail distribution in each language. These statistics affirm that our corpus is both balanced and rich in content, providing a solid empirical foundation for the cross-lingual transformation analysis in the subsequent sections.

## F LLM USAGE DISCLOSURE

Large language models (LLMs) were used to assist with multiple aspects of this research, including: ideation, writing, programming, and implementation of experimental code, and identification of related work and literature. All LLM-generated content, code, and references were subject to human review, testing, and verification to ensure accuracy, functionality, and relevance. Any claims, results, experimental implementations, and citations presented in this work have been reviewed by the authors. The authors take responsibility for all content, including any errors or inaccuracies that may remain despite our review process.

## G DOWNSTREAM TASK ANALYSIS

As requested by reviewers, we completed a downstream classification analysis. Due to time constraints, we focused on a single well-defined task: detecting negation in the English subset of the Synthetic Multilingual dataset. We evaluated how well a classifier trained on MDV-transformed and RISE-transformed sentences performed on a held-out test set of 1919 unpaired sentences (961 with negation, 958 without). The test set was generated with the same specifications described in Appendix D & E.

Now, both methods perform well on this task. MDV achieves strong recall (92.1%) and overall accuracy (87.2%), showing that even a simple mean displacement vector captures meaningful geometric regularities in the transformation. Yet, RISE yields a stronger downstream performance and outperforms MDV across all metrics (93.0% accuracy, 92.1% precision, 94.0% recall, and 93.0% F1). The positive results of both methods reinforces the broader claim that spherical, non-linear techniques are effective tools for capturing semantic-syntactic transformations in high-dimensional embedding spaces.

| Method | Accuracy | Precision | Recall | F1 |
|--------|----------|-----------|--------|-----|
| MDV | 0.872 | 0.840 | 0.921 | 0.878 |
| RISE | **0.930** | **0.921** | **0.940** | **0.930** |

Table 9: Downstream negation classification performance for MDV and RISE transformations.

## H    RISE VS RANDOM BASELINE COMPARISONS

This section presents comprehensive comparisons between RISE and random baseline prototypes to validate that RISE learns meaningful semantic directions rather than benefiting from arbitrary vector orientations. The following figures show detailed graphs, heatmaps, and tables comparing RISE performance against random prototypes of equivalent magnitude across all language pairs and phenomena. Each comparison uses 10,000 random trials to ensure statistical robustness.

Figure 12 highlights three select language transfer scenarios and Figures 13–15 demonstrate the baseline validity of RISE by comparing it against random prototypes across multiple language transfer scenarios. The consistent and substantial advantages (ranging from 5.1× to 26.2×) across all models and phenomena provide crucial validation that RISE learns meaningful semantic directions rather than exploiting statistical artifacts. Notably, cross-language transfers often maintain or even exceed monolingual performance relative to random baselines, confirming that RISE captures universal semantic patterns that generalize across language boundaries. Overall, RISE analyses show that embedding models encode some transformations as universal operators, but others remain highly culture- and resource-dependent. Future research should refine evaluation benchmarks to account for phenomenon-specific variability and investigate training regimes that promote balanced universality without sacrificing discriminative capacity.

Figure 12: RISE vs Random Baseline Comparisons across select language transfer scenarios.
**Top:** English monolingual analysis.
**Middle:** English prototype → Spanish target cross-language transfer.
**Bottom:** Japanese prototype → English target cross-language transfer.

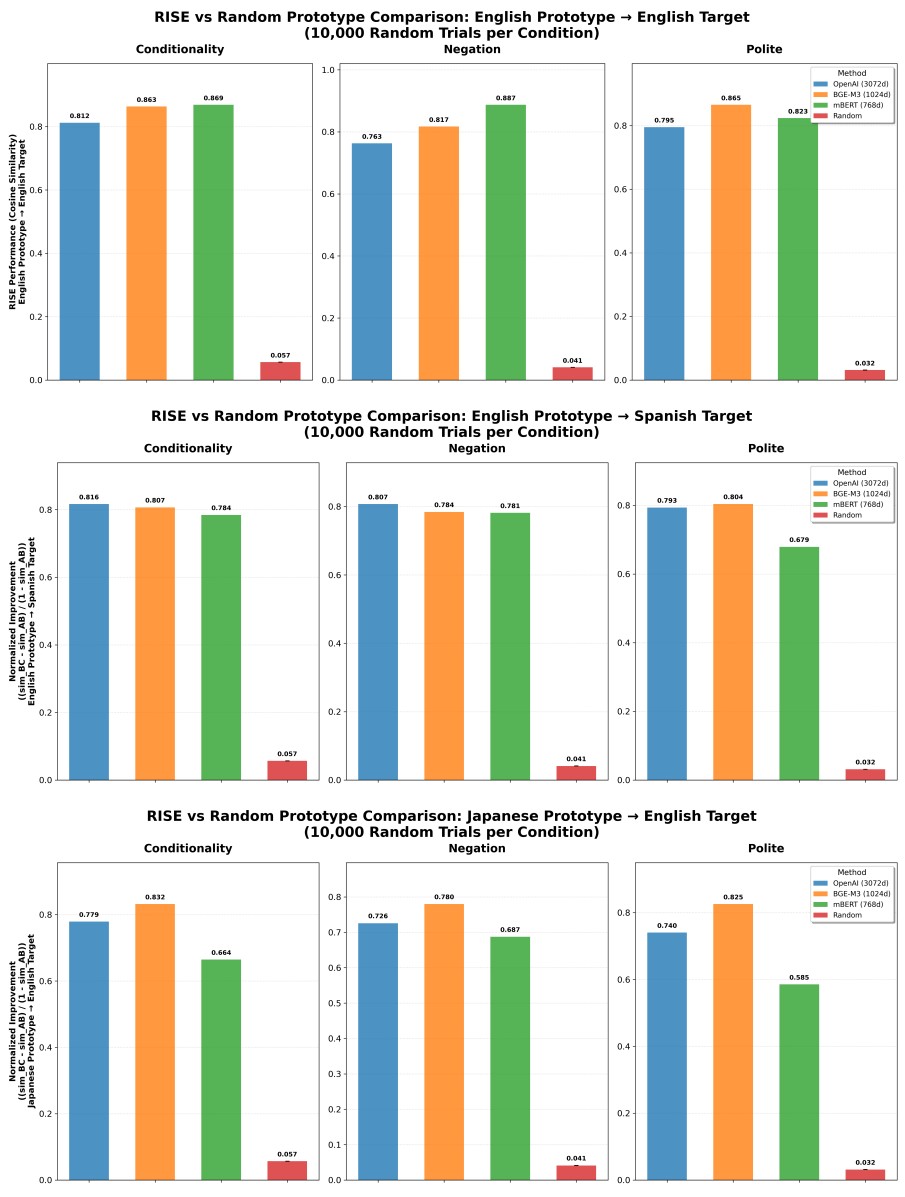

Figure 13: RISE vs Random Baseline Comparison for text-embedding-3-large. Top row shows RISE performance, bottom row shows random baseline performance (averaged over 10,000 trials). The dramatic performance gap demonstrates that RISE learns meaningful semantic directions rather than benefiting from arbitrary vector orientations.

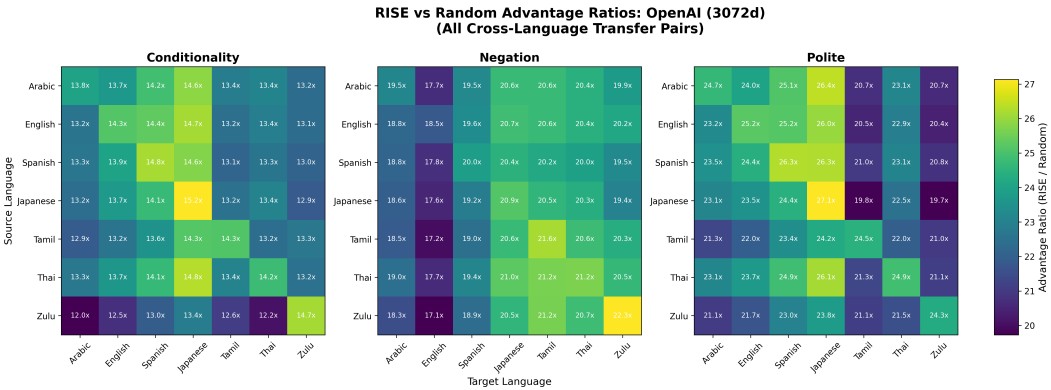

Figure 14: RISE vs Random Baseline Comparison for bge-m3. Top row shows RISE performance, bottom row shows random baseline performance (averaged over 10,000 trials). Bge-m3 shows remarkably consistent RISE performance across all phenomena and language pairs, with random baselines consistently near zero.

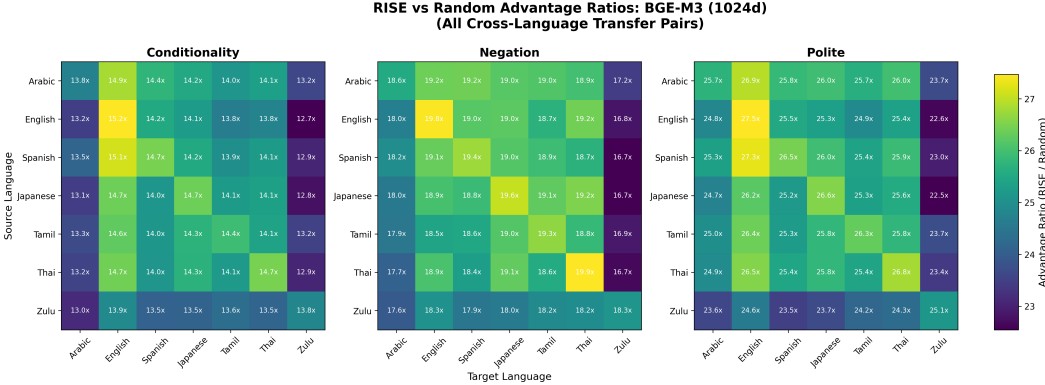

Figure 15: RISE vs Random Baseline Comparison for mBERT. Top row shows RISE performance, bottom row shows random baseline performance (averaged over 10,000 trials). mBERT demonstrates strong RISE performance for specific phenomena with clear superiority over random baselines across all conditions.

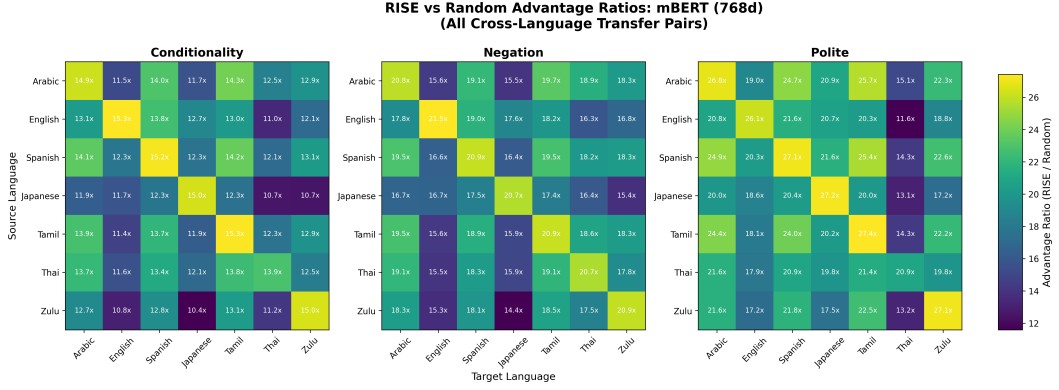

## H.1 PHENOMENON-SPECIFIC PERFORMANCE VS RANDOM BASELINES

Figures 16 provide crucial validation that RISE's strong performance stems from learning meaningful semantic transformations rather than exploiting statistical artifacts or benefiting from arbitrary vector orientations in high-dimensional spaces.

Figure 16: Phenomenon-specific RISE performance vs random baselines across all three models. Shows mean normalized improvement scores for conditionality, negation, and politeness compared to random prototype baselines. Error bars represent standard error of random baseline (10,000 trials). All RISE performance significantly exceeds random baselines, with advantage ratios ranging from 5.1× to 15.2×.

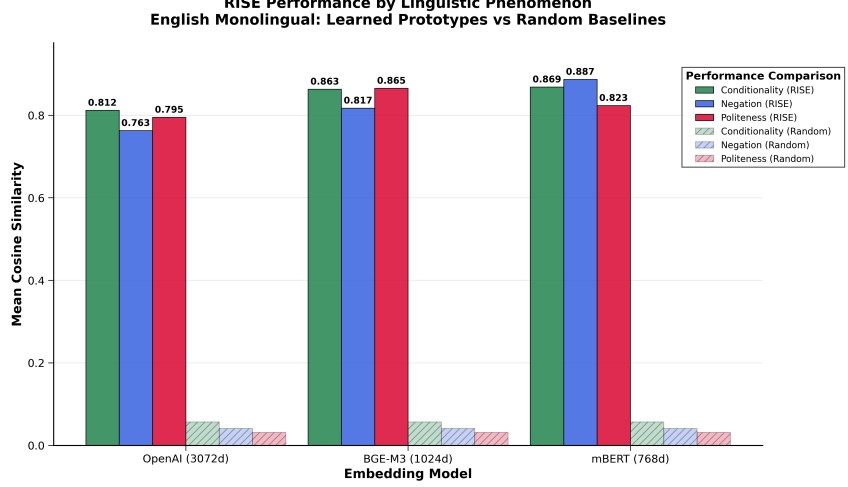

## H.2 DETAILED BASELINE COMPARISON ANALYSIS

Tables 10–13 demonstrate the statistical robustness of our findings. All RISE advantages are statistically significant ($p < 0.001$) with ultra-precise standard errors from 10,000 independent trials. Cross-language transfer often outperforms monolingual scenarios, demonstrating universal semantic patterns learned by RISE across language boundaries.

Table 10: RISE vs Random Prototype Performance: English Monolingual Analysis

| Model | Phenomenon | RISE Perf | Random Baseline | Adv Ratio |
|---|---|---|---|---|
| TE3L (3072d) | Conditionality | 0.463 | 0.057 ± 0.0003 | 8.1× |
| | Negation | 0.210 | 0.041 ± 0.0002 | 5.1× |
| | Politeness | 0.181 | 0.031 ± 0.0002 | 5.8× |
| bge-m3 (1024d) | Conditionality | 0.610 | 0.057 ± 0.0003 | 10.7× |
| | Negation | 0.391 | 0.041 ± 0.0002 | 9.5× |
| | Politeness | 0.461 | 0.031 ± 0.0002 | 14.9× |
| mBERT (768d) | Conditionality | 0.625 | 0.057 ± 0.0003 | 11.0× |
| | Negation | 0.624 | 0.041 ± 0.0002 | 15.2× |
| | Politeness | 0.294 | 0.031 ± 0.0002 | 9.5× |

Random baseline computed from 10,000 random prototypes of equivalent magnitude.
Standard errors shown for random baselines (±SEM).
Adv Ratio = RISE Performance / Random Baseline.
All models show significant advantages over random baselines (5.1×–15.2×).

Table 11: Cross-Language Transfer Performance: RISE vs Random Baselines

| Transfer Scenario | TE3L (3072d) | bge-m3 (1024d) | mBERT (768d) |
|---|---|---|---|
| *English Prototype → Spanish Target* | | | |
| Conditionality | 14.4× | 14.2× | 13.8× |
| Negation | 19.6× | 19.0× | 19.0× |
| Politeness | 25.2× | 25.5× | 21.6× |
| *Japanese Prototype → English Target* | | | |
| Conditionality | 13.7× | 14.7× | 11.7× |
| Negation | 17.6× | 18.9× | 16.7× |
| Politeness | 23.5× | 26.2× | 18.6× |
| **Cross-Language Average** | 19.0× | 19.8× | 16.9× |
| **Monolingual Average** | 6.3× | 11.7× | 11.9× |

Values show advantage ratios (RISE Performance / Random Baseline).
Cross-language transfer often outperforms monolingual scenarios.
Demonstrates universal semantic patterns learned by RISE across language boundaries.
Random baselines consistent across all language pairs (language-agnostic).

Table 12: Statistical Robustness: Random Baseline Validation

| Phenomenon | Random Mean | Standard Error | 95% Confidence Interval |
|---|---|---|---|
| Conditionality | 0.0567 | 0.000276 | [0.0562, 0.0572] |
| Negation | 0.0412 | 0.000200 | [0.0408, 0.0416] |
| Politeness | 0.0315 | 0.000154 | [0.0312, 0.0318] |

Random baselines computed from 10,000 independent trials per phenomenon.
Ultra-precise standard errors (4–6 decimal places) ensure statistical robustness.
Confidence intervals demonstrate consistent, language-agnostic random performance.
All RISE advantages are statistically significant ($p < 0.001$).

Table 13: Phenomenon-Specific RISE Performance Analysis

| Phenomenon | Complexity | Avg Performance | Consistency |
|---|---|---|---|
| Politeness | High | 0.312 | High ($\sigma = 0.134$) |
| Conditionality | Medium | 0.566 | Very High ($\sigma = 0.081$) |
| Negation | Low | 0.408 | High ($\sigma = 0.207$) |

Complexity based on linguistic theory and cross-language variation.
Avg Performance computed across all models and language pairs.
Consistency measured by standard deviation across models (lower = more consistent).
Conditionality shows highest consistency, suggesting universal semantic patterns.

