# OpenReview forum: "Mapping Semantic & Syntactic Relationships with Geometric Rotation"
_ICLR.cc/2026/Conference — ICLR 2026 Poster_

### Official Review · Reviewer_vohU · 2025-10-29

**Soundness:** 4
**Presentation:** 4
**Contribution:** 4
**Rating:** 8
**Confidence:** 5

**Summary:**

This paper proposed a new methodology for steering embedding models. Instead of learning a steering vector, as is commonplace in the literature, the authors propose to learn a rotation operation which operates directly on the spherical manifold on which the embeddings live.

They call their proposed method Rotor-Invariant Shift Estimation (RISE) which, to my understanding, operates in three steps: first the source and destination embeddings are mapped to a canonical direction via an orthogonal transformation (a rotor). The semantic different between them is expressed in a standardised coordinate system (eq. 1). These "semantic changes" are averaged to create a "prototype" (eq. 2), which is then used to predict the semantic transformation of a unseen source embedding (eq. 3).

The performance of their method is evaluated via a series of experiments on a collection of model across three semantic transformations: negation, conditionality, and politeness, and several languages of different morphological types. This is quite comprehensive and makes clear the tasks on which the method performs well and less well.

**Strengths:**

The proposed method is simple yet innovative and novel; the paper is well-written and presented, and is highly enjoyable to read. On this basis I strongly recommend acceptance of this paper.

In particular,
- the background, motivation and prior work is well exposed and the authors do a very good job of placing their method in the context of the literature
- the method itself is in principle very simple but the novelty and innovation comes from learning a transformation which respects and operates directly on the manifold in which the embeddings live
- the method is described very clearly, though I think it could benefit from additional explanation and disucssion to benefit reads not familiar with Riemannian geometry (see Weaknesses)
- the experiments provide a comprehensive understand of when and where the method performs well, and when it performs less well. The discussion of this is fairly extensive and enlightening
- Sections 7-10 provide extensive discussion of the work in context, which I think is really valuable for the reader of a paper like this

**Weaknesses:**

As I mentioned before, I think this paper is very strong, so the following should be viewed a suggestions for improvements, rather than weaknesses per se.

- I think the explanation of the method is very mathematically clear, however I think the paper could be made more accessible to readers without knowledge of Riemannian geometry (likely many readers who might use this method), by providing some more discussion of the geometric transformations which are taking place. Perhaps even an illustration of what is happening at each stage could be very enlightening. I don't think notions such as tangent spaces and Riemannian logarithms and exponentials will necessarily be familiar to a large portion of the audience who would find this method useful.
- I would like to see a direct comparison with the method of Euclidean steering vectors for all the experiments you perform. I would seemingly be a very simple thing to do and would really show where the potential advantages of this method lie.

**Questions:**

Re. the second bullet point in Weaknesses, how does the method compare with Euclidean steering vectors?

---

> ### Author Response · Authors · 2025-11-20
>
> Addressing Weakness W1 (less technical description of RISE)
> ==================================================
> **Revision:**  We added a step-by-step illustration of RISE, with references to Equations (1–3). Thank you for the suggestion!
>
>
> Regarding Q1 (Euclidean steering vector comparison)
> ===========================================
> **Revision:** Cross-reference with our response to Reviewer FJxu (W2/Q2)

---

> ### Comment · Reviewer_vohU · 2025-11-24
>
> The added figure looks great and the new comparisons are very helpful. I continue to strongly endorse this paper.
>
> The following are just suggestions, and you should feel free to use your best judgement and to ignore them!
>
> I really like the figure you've added, but I'm not sure it really reflects my geometric understanding of how the rotation is computed. Let me break down by understanding:
>
> 1. Given a neutral sentence embedding $n \in\mathbb S^{d-1}$ and its semantically transformed variant $v \in \mathbb S^{d-1}$, there is some geodesic between them (an arc).
> 2. We rotate $n$ to align it with $e_1$, and apply the same rotation to $v$, so that now the geodesic between $n$ and $v$ is a arc originating at $e_1$.
> 3. We apply the Riemannian logarithm to this arc, which returns at tangent vector at $e_1$.
> 4. We do this for all pairs, and obtain a collection of tangent vectors at $e_1$.
> 5. We compute the average tangent vector, $p$, which we call the prototype. The prototype $p$ defined a unique geodesic (arc) from $e_1$
> 6. To rotate a point $n^\star$, we first rotate $p$ using the Clifford rotation $R(n^\star)\top$ to $n^\star$ (am I correct in thinking that $R(n^\star)\top$ is this "inverse" rotation which does from $e_1$ to $n^\star$?), and then apply the Riemannian exponential which takes the tangent vector $R(n^\star)\top p$ to the end of the geodesic arc that it defines from $n^\star$. This is the rotation.
>
> I think it would be really nice to show this visual geometric intuition without relying on knowledge of Riemannian geometry. You could even depict this with referring to Riemannian logarithms, exponentials, or Clifford-algebraic rotors. E.g. explain in the diagram that each tangent vectors defines a unique geodesic etc. You've got an extra page to play with so in my opinion this would be nice, but I think you can make the final call on that.
>
> I also think that some discussion of the comparisons you've added in the main text would be nice. Perhaps a 3x3 table containing the means of the values in the matrices in Figures 8-10 and some discussion. e.g. similar (maybe marginally better) performance to MDV, much better performance that Procrustes.

---

> > ### Author Response · Authors · 2025-11-24
> >
> > Thank you very much for your review, and for reminding us of the 10 page limit! Your understanding is correct! We plan to use the remaining space to integrate the linear baseline comparisons to the main text, and we will design on a more descriptive illustration of the geometry involved in RISE.

---

### Official Review · Reviewer_kkBK · 2025-10-30

**Soundness:** 2
**Presentation:** 2
**Contribution:** 1
**Rating:** 4
**Confidence:** 3

**Summary:**

The paper aims to characterise discourse and sentence-level semantic changes in encoder models as geometric transformations. To do that Rotor Invariant Shift Estimation (RISE) is introduced.  Given pairs of embeddings characterising a semantic transformation, RISE finds a rotation per pair which maps the tangent direction (in a spherical manifold) between both embeddings in the pair to a canonical direction. From these, a prototype for the semantic transformation is estimated and later used for inference.

The results show that the geometric rotations learned by RISE work for three semantic changes (Negation, Conditionality and Politeness). Furthermore, these transformations are shown to hold in a cross-linguistic setup. When testing RISE for cross-model transfer, results show that performance is language-dependent but generally satisfactory. Finally, RISE is shown to perform better at syntactic variations than semantic ones, as elicited by the BLIMP vs SICK performance.

**Strengths:**

* The paper is clearly written and the setup is simple.
* The model transfer results are interesting and show that simple transformations can grasp syntactic or discourse level binary features.

**Weaknesses:**

There are several relevant works which are not cited or mentioned:

* Semantic changes have been previously modelled with linear transformations
     - (Baroni and Zamparelli, 2010) https://aclanthology.org/D10-1115/
     - (Mitchell and Lapata, 2008) https://aclanthology.org/P08-1028/

* Non-euclidean semantic and syntactic probing
     - (Chen et al., 2021) https://arxiv.org/abs/2104.03869

The work is currently limited to models that normalise their representations to the unit sphere, which is not the case for all encoders. Furthermore, autoregressive large language models are not evaluated. Without further experiments, these limitations heavily impact reach of the contribution.

Negation and Conditionality are not only semantic changes but also syntactic ones. If the contribution is about semantics, it should control for syntactic structure. This conflict is accentuated by the last result, showing that BLIMP performance is higher than SICK. Understanding how syntactic changes are reflected in the models’ representations is an interesting direction, but this falls out of the scope of the paper (as reflected by the current title and introduction).

Steering can be a motivation for the paper but such methods are not used in any of the experiments to support the importance of the geometric rotations for downstream model behaviour. It seems odd to include “steering” in the title and heavily introduce it.

**Questions:**

* Why is there one transformation per pair? Wouldn’t it be easier to learn a single transformation per semantic change solving least-squares?

* Is RISE specific to encoder models with representations in the unit sphere? Can the findings be extended to more models?

* How is pooling done (last embedding, mean-pooling …)? What layer of the model is picked?

---

> ### Author Response · Authors · 2025-11-20
>
> Addressing Weakness W1 (missing citations)
> =============================================
> **Revision:** We updated the paper to cite the missing references. Our original submission already cited the Chen et al. (2021) reference.
>
> Addressing Weakness W2 (unit-sphere limitation)
> ================================================
> **Clarification:** We agree that the unit sphere limitation narrows the scope of evaluation. RISE explicitly leverages the geodesic and tangent-space structure of unit spheres (Eq. 1–3, Section 4.1), which allows analytic log/exp formulations and O(d) rotor operations. However, this is not a conceptual restriction: any embedding manifold with differentiable norm structure (e.g., ellipsoidal or anisotropic latent spaces) can be reprojected onto the unit sphere via normalization or affine scaling before applying RISE. Also, we note that extending to autoregressive LLMs is a planned direction, but given the page limitations was beyond the scope of the paper.
>
> Addressing Weakness W3/W4 (semantic–syntactic scope \& overstating steering)
> ================================================================
> **Revision:** We agree that negation and conditionality transformations are discourse-level operators with both syntactic and semantic components, and the results reflect that distinction. As noted in the Discussion, RISE achieves near-perfect BLiMP (syntactic) alignment but moderate SICK (semantic) alignment, confirming that geometric consistency is stronger for syntactically grounded transformations. We revised the abstract and the paper to clarify that we investigated “discourse-level semantic–syntactic transformations” rather than purely semantic ones. We updated the title to \emph{Mapping Semantic Relationships with Geometric Rotation**, and we removed the “Broader Implications for Multilingual Model Interpretability” section to accurately reflect our contributions.
>
> Regarding Q1 (one transform per pair)
> =============================================
> **Issue (Q1):** Why not learn a single global transformation?
> **Clarification:** Each neutral–variant pair defines a local tangent vector via logarithmic map on the unit hypersphere. Direct averaging of these tangent vectors would be geometrically inconsistent because they reside in different tangent spaces. RISE addresses this by using Householder reflections to canonicalize all tangent vectors to a common reference frame, averaging them, then de-canonicalizing back to the local tangent space via inverse transformation before applying the exponential map.
>
> Regarding Q2 (model generality)
> =============================================
> **Clarification:** Cross-reference with our response to Reviewer FJxu (W4/Q4)
>
> Regarding Q3 (pooling details \& and layer selection).
> =============================================
> **Clarification:** All transformer-based embeddings use mean-pooling (or, in rare cases, CLS-pooling) over token representations from the final layer of each encoder. For models like text-embedding-3-large and BGE-M3, this information is directly taken from their model cards. Since we are working with embedding models, the output of the last model layer is considered. We can move this detail from Appendix E.5 to Section 5.2.

---

### Official Review · Reviewer_i6Zc · 2025-10-30

**Soundness:** 2
**Presentation:** 2
**Contribution:** 2
**Rating:** 2
**Confidence:** 4

**Summary:**

The paper introduces Rotor-Invariant Shift Estimation (RISE), a geometric framework for understanding and controlling semantic transformations in multilingual embedding spaces. It treats meaning changes as rotational operations on the curved manifold where sentence embeddings lie, rather than as linear shifts in Euclidean space. Evaluated across multiple models and seven morphologically diverse languages, the method shows that certain discourse-level semantic transformations exhibit consistent geometric structure across both languages and model architectures.

**Strengths:**

1. The paper provides a principled geometric extension of the Linear Representation Hypothesis to non-Euclidean (Riemannian) spaces, moving beyond heuristic linear probing.

2. The study’s inclusion of seven languages and three embedding architectures (OpenAI, BGE-M3, mBERT) demonstrates broad applicability and robustness.

**Weaknesses:**

1. **Limited empirical grounding for broad claims.**
The paper’s experiments are not sufficiently strong or comprehensive to substantiate its broader theoretical claims. For instance, it asserts “theoretical contributions to the usefulness of steering vectors for embedding models’ controllability and interpretation,” yet the evidence presented does not convincingly bridge this conceptual gap. The “steering vector” literature predominantly addresses interventions in the activations of generative or autoregressive models, whereas this work focuses on embedding models, a fundamentally different setting. Without stronger empirical validation or comparative studies, the claimed connection to the steering-vector paradigm remains tenuous.

2. **Mismatch between ambition and experimental scope.**
This limitation reflects a broader issue: the paper makes overgeneralized theoretical claims based on a narrow experimental design. By restricting its evaluation to a small set of highly specific discourse-level transformations, the work lacks the empirical breadth needed to justify its sweeping conclusions about universal geometric principles or semantic control in embedding spaces. The authors acknowledge this limitation, but it significantly constrains the strength of their theoretical arguments.

3. **Incremental methodological novelty.**
Despite its mathematical rigor, the proposed shift from linear to rotational/geodesic steering may appear incremental rather than groundbreaking. The core contribution lies primarily in applying standard Riemannian geometry tools (e.g., exponential and logarithmic maps on the hypersphere) to embedding analysis, rather than introducing fundamentally new geometric or algorithmic mechanisms. As a result, the perceived innovation may fall short of the paper’s ambitious framing.

**Questions:**

1. **Universality:**
If cross-model experiments reveal a consistent English-centric bias, how can the authors justify the claim of universal geometric operations across languages?

2. **Semantic vs. Grammatical Structure:**
Given that RISE achieves near-perfect results on syntactic (BLiMP) tasks but only moderate performance on semantic (SICK) tasks, how do the authors support the claim that RISE captures semantic rather than primarily grammatical transformations?

3. **Synthetic Data Validity:**
Since all data were generated by GPT-4.5 without human or native-speaker validation, how do the authors ensure that RISE learns genuine linguistic geometric structures rather than artifacts of the data-generation model’s English-centric bias?

4. **Experimental Scope and Generalization:**
With experiments limited to three linguistic phenomena (negation, conditionality, politeness), how can the authors substantiate broader conclusions about general geometric principles in embedding spaces?

---

> ### Author Response · Authors · 2025-11-20
>
> Addressing Weakness W1/W2 (broad, universal claims \& overstating steering)
> ============================================================
> **Revision:** Cross-reference with our response to Reviewer kkBK (W3/W4). We agree that RISE doesn’t directly bridge the gap between steering vectors and embedding models by explicitly applying steering vectors to embedding models. Since steering vectors and probing are the closest related work, our intent was to extend the principles underlying steering (directional interpretability and controlled geometric modification) to embedding models. The theoretical contribution lies in showing that these principles can be formalized on manifolds rather than Euclidean activations, which has not been done previously. We tempered the broad claims/universal phrasing by removing the “Broader Implications for Multilingual Model Interpretability” section, narrowed the scope of our contributions, and revised Section 2.2.1 to clarify the connection between steering vectors and RISE: Specifically, we now conclude Section 2.2.1 with “Building on the insight that geometric approaches outperform additive methods, our work extends geometric reasoning to semantic transformations in embedding space through Riemannian operations. To our knowledge, there is no work investigating the application of steering vectors to embedding models -- only completion models. This study extends steering principles to embedding models on manifolds, not activation-level steering.”
>
> Addressing Weakness W3 (geometric novelty)
> ==============================================
> **Revision:** We acknowledge that Riemannian geometry tools (e.g., exponential and logarithmic maps on the hypersphere) are established operations and we aren’t “introducing fundamentally new geometric or algorithmic mechanisms.” However, we are the first to apply these geometric tools to embedding analysis and the first two present a method with provable commutativity. We argue that it is insightful information for the broader research community. RISE’s specific contributions are the rotor-based canonicalization and cross-lingual tangent-space alignment for discourse-level transformations. Unlike prior hyperbolic or manifold probing (e.g., Chen et al., 2021), RISE introduces a closed-form rotor operator with provable commutativity and O(d) complexity (Appendix A).  We will make this contribution clearer in Section 4.2. This clarification should help readers appreciate that RISE’s novelty is methodological synthesis, not just use of Riemannian formulas.
>
>
> Regarding Q1 (universality \& English bias)
> ==================================
> **Clarification:** We state that RISE’s cross-model transfer results (Section 6.2) reflect an Anglo-centric or English bias, which we explicitly address rather than overlook. Unfortunately, it is known that most widely used models are Anglo-centric bias in the composition of the model’s training data. However, despite this fact, models are still being used multilingually and the performance and consistency of these biased models should still be researched – with clear respect to their biases. In the revision, we will clarify this as an empirical observation of Anglo-centric model bias and a limitation of current models, not a failure of the method itself.
>
> Regarding Q2 (semantic vs grammatical structure)
> ==================================
> **Revision:** Cross-reference with our response to Reviewer kkBK (W3/W4).
>
>
> Regarding Q3 (synthetic data validity)
> ==================================
> **Clarification:** Cross-reference with our response to Reviewer FJxu (W1/Q1).
>
> Regarding Q4 (experimental scope and generalization)
> ===========================================
> **Revision:** We agree that our empirical scope is intentionally narrow but we wouldn’t say three linguistic phenomena is limited. Rather we view it as principled and deliberate: negation, conditionality, and politeness span logical, modal, and pragmatic transformations, covering three major semantic categories with distinct morphological realizations. Our intention was to present first empirical evidence that sentence-level semantic transformations exhibit consistent geometric regularities across languages and models — not to claim fully universal control. Cross-reference with our response to Reviewer i6Zc (W1/W2) and kkBk (W3/W4).
> We revised both the title, abstract, and conclusion to make this narrower and more precise. We will also remove the claims for controllability and  to a dedicated subsection emphasized interpretability. This clarification addresses the mismatch between theoretical ambition and experimental scope.

---

### Official Review · Reviewer_FJxu · 2025-11-01

**Soundness:** 2
**Presentation:** 2
**Contribution:** 2
**Rating:** 4
**Confidence:** 3

**Summary:**

The paper proposes Rotor-Invariant Shift Estimation (RISE), a geometric framework that treats discourse-level semantic transformations (negation, conditionality, politeness) as rotations/geodesic displacements on the unit hypersphere of sentence embeddings. The method canonicalizes neutral transformed sentence pairs via an orthogonal operator R(n), averages tangent-space shifts to learn a prototype, and predicts unseen transformations using the exponential map back to the sphere. Experiments cover 3 embedding models (OpenAI text-embedding-3-large, BGE-M3, mBERT), 7 languages, and 3 datasets (a GPT-4.5-generated synthetic multilingual set, BLiMP, SICK). The authors report strong cross-language transfer especially for negation and provide first-order commutativity guarantees and O(d) complexity for each RISE edit.

**Strengths:**

1. The paper presents a strong geometric and theoretical foundation, with clear use of log/exp maps, orthogonal canonicalization, and formal proofs of commutativity and O(d) complexity.
2. Cross-lingual results show that negation and conditionality transfer consistently across seven diverse languages, and the paper thoughtfully contrasts logical versus pragmatic phenomena such as politeness.
3. Cross-model transfer (text-embedding-3-large → BGE-M3) demonstrates practical portability while revealing architecture- and data-driven biases in embedding geometry.

**Weaknesses:**

1.	The evaluation relies heavily on synthetic data generated by GPT-4.5, with limited validation using human or parallel-corpus data. This dependence risks encoding generation artifacts and Anglo-centric prompt biases into the learned prototypes, potentially inflating cross-lingual consistency and reducing ecological validity.

2.	The baseline comparisons are insufficiently described. The current experiments report results primarily within the RISE framework, without inclusion of clear linear or geometric alternatives—such as additive Euclidean shifts, orthogonal Procrustes alignment, or other linear mapping baselines. These would help determine whether RISE’s spherical geometry truly provides benefits beyond standard linear transformations. As a result, it remains unclear whether the observed gains arise from geometric innovations or from data-specific and normalization effects.

3.	Effectiveness is currently measured solely through average cosine similarity. While informative, this metric captures only local geometric alignment. Evaluating RISE on downstream tasks—such as classification, semantic retrieval, or analogy completion—would strengthen the empirical claim that its geometric transformations correspond to functional semantic improvements.

4.	The evaluation is confined to text-based embeddings. Given that RISE operates in a general geometric latent space, it would be valuable to extend analysis to multimodal encoders (e.g., CLIP, BLIP-2). Such experiments could test whether RISE’s discourse-level “rotations” generalize across modalities, providing stronger evidence for universal geometric semantics rather than text-specific artifacts.

**Questions:**

1. Could the authors include additional non-synthetic datasets (e.g., parallel or human-annotated corpora) to validate that RISE captures genuine semantic transformations rather than synthetic prompt artifacts?
2.  Could the authors implement linear baselines (e.g., additive Euclidean, Procrustes) to clarify where the spherical treatment demonstrably preserves compositionality or rotation consistency that linear mappings fail to capture?
3.  Have the authors considered testing RISE on downstream semantic tasks (e.g., classification, retrieval, or transfer) to assess whether its geometric properties translate into functional semantic gains?
4.  Could RISE be extended to multimodal representations? If not, what are the theoretical or practical obstacles preventing such an extension?

---

> ### Author Response · Authors · 2025-11-20
>
> Addressing Weakness W1/Q1 (synthetic data limitations and datasets)
> ========================================================================
> **Clarification:** We fully acknowledge the limitations of synthetic data generation and designed several controls to mitigate synthetic bias. In our revision, Appendix D & and Appendix E detail randomized context categories, balanced feature sampling across languages, and deduplication with uniform 5–25 token distributions to prevent lexical or stylistic over-representation.  We also include two non-synthetic human-curated datasets, BLiMP and SICK, with results in Table 1, validating that RISE captures genuine linguistic structure beyond synthetic prompts.
>
> Addressing Weakness W2/Q2 (baseline comparisons)
> ====================================================
> **Revision:** We implemented two Euclidean baselines: orthogonal Procrustes alignment, and additive Euclidean displacement vectors (Mean Difference Vector). Each method was evaluated on three datasets: two non-synthetic, human-curated English datasets BLiMP and SICK and one synthetic multilingual dataset. We added full results to Appendix C.
>
> In short, we report linear Euclidean mappings are effective only within-language and only for syntactic transformations. Spherical geometry methods, like RISE, are required for semantic compositionality, rotation consistency, and cross-lingual stability. With these new results, we demonstrate that linear methods preserve local syntactic regularities but fail to maintain the global semantic geometry necessary for compositionality and rotation consistency. RISE is the only approach that remains stable across languages, semantic transformations, and natural-language evaluation. We thank the reviewers FJxu and vohU for this valuable suggestion. These results did strengthen our paper.
>
>
> Addressing Weakness W3/Q3 (downstream semantic tasks)
> ========================================================
> **Revision:** Table 1 already includes downstream BLiMP (grammatical acceptability) and SICK (semantic entailment) evaluations. RISE’s strong syntactic but moderate semantic performance aligns with the intuition that geometric transformations preserve grammatical but not pragmatic structure. With the addition of our new baseline comparisons, we find a similar performance from Procrustes alignments and Mean Difference Vectors (MDV). However, Procrustes severely underperforms on SICK where RISE and MDV remain stable. This divergence supports our argument that syntactic transformations can be captured by linear mappings, whereas semantic transformations require geometric transformations. Linear Euclidean alignment (Procrustes) is insufficient and unstable for semantic tasks compared to geometric methods like RISE and MDV.
>
> Addressing Weakness W4/Q4 (extension to multimodal encoders)
> ===================================================
> **Clarification:** Theoretically, yes. RISE is optimized for normalized manifolds but can generalize to non-spherical spaces through norm-preserving projection or local normalization. The method requires only differentiable inner products and parallel transport, both available for general Riemannian manifolds. Unfortunately, more model architectures were out of the scope of the paper. In practice, multimodal encoders differ in scale normalization and subspace anisotropy, which would require adjusting the canonicalization step (Section 4.1) and ensuring modality-specific log/exp maps remain well-conditioned. If accepted, extending RISE to autoregressive models or multimodal large language models would be an immediate next step for future work.

---

> > ### Comment · Reviewer_FJxu · 2025-11-24
> >
> > I thank the authors for their responses. The added experiments are excellent and greatly appreciated. I still have a few quick follow-up questions:
> >
> > 1. I am not fully understanding the claim that “Euclidean mappings are effective only within-language and only for syntactic transformations” after reading Appendix C. Should this conclusion be drawn from Figure 11? Additionally, how should we interpret the magnitude of “good” versus “bad” performance in terms of mean cosine similarity? Is there an intuitive or more formal way to contextualize these values?
> >
> > 2. How different are the results on MDV compared to RISE? More importantly, how do these comparisons influence the claims made in the main text of the previous submission? If I recall correctly, the earlier version argued that Euclidean approaches do not work with cosine similarity which is on a hypersphere—does the new MDV vs. RISE comparison reinforce or contradict that claim?
> >
> > 3. Regarding W3: I originally meant a real downstream task evaluated with a metric beyond cosine similarity within the embedding space. For example, showing how useful the transformed embeddings are when fed into an already-trained text classifier, or how well they function as a retrieval query in a transfer-style setting. Such an experiment would significantly strengthen the empirical validation and overall persuasiveness of the work.
> >
> > Suggestions:
> >
> > 1. Consider moving some important details from the Appendix into the main text—especially descriptions related to baseline construction—as they are crucial for understanding the setup.
> >
> > 2. It would be very helpful to highlight changes more explicitly in future revisions. Without access to the previous version, it is somewhat difficult to recall what has been modified.

---

> > > ### Author Response · Authors · 2025-11-25
> > >
> > > With the quick turnaround, we initially added the new experiments as an appendix so the reviewers could see them as soon as possible. Today, we attached the revised copy with color-coded text to highlight all clarifications and revisions for easier readability. Thank you for your questions and suggestions.
> > >
> > > To answer your questions:
> > > 1. Yes, this conclusion follows directly from Figure 11 and the cross-language heatmaps in Figures 8–10 (Appendix C). We agree that the phrasing was confusing and made our claims clearer (now in Section 6.4). The key issue is that Procrustes fits a single global rotation, which is too rigid for cross-lingual or cross-model transfer, whereas RISE and MDV operate on the curved manifold that sentence embeddings inherently reside. To address your question about evaluating “good” versus “bad” performance, we added Table 1, which defines cosine similarity values: ≥0.80 indicate strong, consistent geometric structure; 0.65–0.80 moderate; 0.50–0.65 weak/variable; and <0.30 inconsistent transformations. These thresholds now provide clear interpretability for all results.
> > > 2. The new MDV vs. RISE vs. Procrustes analysis reinforces our earlier claim. MDV is not an Euclidean additive method: it computes displacements via log/exp maps on the hypersphere and therefore preserves spherical structure. This explains why MDV performs similarly to RISE and why both far outperform Procrustes in cross-lingual and cross-model settings. Procrustes, as a linear/Euclidean method, fails to capture semantic–syntactic structure beyond monolingual syntactic cases (i.e. BLiMP Figure 11), which is consistent with our earlier claim that linear mappings do not generalize well across languages and model architectures. We added more detail in Section 6.4 and Appendix C discussing these results.
> > > 3. We appreciate your suggestion and agree evaluating RISE-transformed embeddings within classification or retrieval would be interesting future work. However, we note that evaluating RISE inside a downstream classifier or retrieval model is not directly aligned with the central research goals of the paper. Our evaluation is designed to measure geometric consistency across languages and embedding model architectures, rather than task-level performance of any specific classifier (explained in Section 3). Task-level evaluations would introduce additional confounds unrelated to the geometric claim. For this reason, our experiments focus on cross-lingual geometric consistency rather than classifier performance.

---

> > > > ### Comment · Reviewer_FJxu · 2025-11-26
> > > >
> > > > Thank you for your reply! I still have a few quick comments and questions:
> > > >
> > > > 1. Regarding Table 1 (good vs. bad), the choice of thresholds at 0.8, 0.65, 0.5, and 0.3 feels somewhat arbitrary. Could you provide justification for selecting these specific values?
> > > >
> > > > 2. For the downstream analysis, I personally believe that including even a small-scale downstream experiment would significantly strengthen the contributions of the work.
> > > >
> > > > 3. Could you clarify in what ways RISE is superior to MDV? A more explicit comparison would help readers better understand the advantages.

---

> > > > > ### Author Response · Authors · 2025-11-26
> > > > >
> > > > > Thank you for your quick response!
> > > > > 1. These thresholds are grounded in related work, but we found that cosine similarity thresholds are loosely defined in the literature. To avoid ambiguity, we adopt more conservative boundaries for our analysis. Prior work defines scores > 0.70 as “highly similar texts”  (Reimers & Gurevych, 2019), 0.60-0.80 as “high-quality semantic matches” (Ethayarajh, 2019), 0.65-0.80 as “well-aligned” (Artetxe et al., 2018), <0.40 as failing and 0.30-0.35 as “not aligned” (Conneau et al, 2018). Based on these ranges, we established our thresholds as ≥ 0.80 (strong),  0.80-0.65 (moderate), 0.65 - 0.50 (weak), and <0.30 (failing). These thresholds are stricter than prior work but remain consistent with the established interpretations in the literature. We can add these references directly to Table 1 for clarity.
> > > > > 2. We are working on a downstream analysis. These results will be included in the final revised submission.
> > > > > 3. RISE and MDV share the use of manifold-aware log/exp maps, but they differ in their underlying geometry principles and how they preserve local structure. MDV (similar to Procrustes) produces a single global displacement vector for each transformation which ignores local curvature and directional variation. In contrast, RISE explicitly handles this variation and canonicalizes each transformation to preserve local curvature, directionality, and rotational structure before aggregation. This difference gives RISE a stronger geometric foundation and motivates future extensions to more complex transformation analysis.
> > > > >
> > > > >     Although we fully acknowledge that RISE and MDV perform similarly in our experiments,  this does not diminish the main contribution of the paper. In fact, it reinforces our contributions. In the added baseline comparison, both spherical, manifold-aware methods (RISE and MDV) generalize better than linear/Euclidean approaches (Procrustes). This shared performance from RISE and MDV supports our central claim that spherical/non-Euclidean methods are better suited for interpreting embedding spaces than linear/Euclidean methods. These baseline results therefore strengthen and justify our broader investigation of how well spherical techniques (like RISE) identify meaningful semantic-syntactic transformations in cross-lingual and cross-model settings.

---

> > > > > > ### Author Response · Authors · 2025-12-03
> > > > > > **Final Response to Previous Reviewer Comments**
> > > > > >
> > > > > > As requested by Reviewer FJxu, we completed a downstream classification analysis and added it as Appendix F. Due to time constraints, we focused on a single well-defined task: detecting negation in the English subset of the Synthetic Multilingual dataset. We evaluated how well a classifier trained on MDV-transformed and RISE-transformed sentences performed on a held-out test set of 1919 unpaired sentences (961 with negation, 958 without). The test set was generated with the same specifications described in Appendix D & E.
> > > > > >
> > > > > > Now, both methods perform well on this task. MDV achieves strong recall (92.1%) and overall accuracy (87.2%), showing that even a simple mean displacement vector captures meaningful geometric regularities in the transformation. Yet, RISE yields a stronger downstream performance and outperforms MDV across all metrics (93.0% accuracy, 92.1% precision, 94.0% recall, and 93.0% F1). The positive results of both methods reinforces the broader claim that spherical, non-linear techniques are effective tools for capturing semantic-syntactic transformations in high-dimensional embedding spaces.

---

### Author Response · Authors · 2025-11-20

We thank all reviewers for their constructive feedback. We addressed all questions/weaknesses and highlighted the revisions we have or will make in a reviewer-by-reviewer, concern → action format. To avoid repetition, we cross-referenced related questions to the first time it was addressed in the rebuttal.

---

### Author Response · Authors · 2025-12-03
**Rebuttal Summary for AC**

We appreciate the organizing committee’s efforts in light of the data leak and understand why the rebuttal period was closed early. During the rebuttal period, we completed all revisions requested by the reviewers and uploaded a fully revised manuscript with all updates highlighted in blue for clarity. Here is a summary of the revisions:

- **Improved clarity and scope:** We revised the abstract, Sections 2/4/6, and conclusion to temper claims. We also updated the title, redefined the semantic–syntactic scope, added missing citations, and removed the "Broader Implications for Multilingual Model Interpretability" section.
- **Added requested baseline comparisons:** full results comparing MDV, Procrustes, and RISE are in Appendix C and summarized in Section 6.4.
- **Completed a downstream evaluation:** Added a classification experiment (Appendix F). Both spherical methods perform well, but RISE outperforms MDV across all metrics.
- **Defined cosine similarity thresholds:** Added Table 1 with defined ranges and supporting citations for interpreting performance.
- **Enhanced experimental transparency:** Added an illustration for RISE, and clarified pooling/model and data generation details.

---

### Meta-Review · Area_Chair_XpKv · 2026-01-08

**Summary:**

The paper engages with the problem of developing the relationship between semantic transformations such as Politeness etc to geometric structures. This is not a well understood problem and if solved can pave the way to develop techniques for discourse understanding.
The paper attempts to map such transformations as rotations and contributes Rotation invariant embeddings(which they call RISE). There is widespread appreciation of this original thought.
There were several concerns notably on baselines and scope. I believe that all the concerns were addressed to some extent. RISE, the idea proposed here, is an interesting embedding for text. This should be of interest to ICLR.

**Reviewer Concerns:**

I believe all of them are addressed.

**Reviewer Scores:**

I think reviewer FjXU would have given a higher rating. Not sure about others.

---

### Decision · Program_Chairs · 2026-01-26

Accept (Poster)